# Tailored topotactic chemistry unlocks heterostructures of magnetic intercalation compounds

Samra Husremović [1], Oscar Gonzalez[1], Berit H. Goodge [1,2], Lilia S. Xie[1], Zhizhi Kong[1], Wanlin Zhang[1], Sae Hee Ryu [3], Stephanie M. Ribet[4], Shannon S. Fender [1], Karen C. Bustillo [4], Chengyu Song[4], Jim Ciston [4], Takashi Taniguchi [5], Kenji Watanabe [6], Colin Ophus [7], Chris Jozwiak [3], Aaron Bostwick [3], Eli Rotenberg [3] & D. Kwabena Bediako [1,8,9] ✉

The construction of thin film heterostructures has been a widely successful archetype for fabricating materials with emergent physical properties. This strategy is of particular importance for the design of multilayer magnetic architectures in which direct interfacial spin-spin interactions between magnetic phases in dissimilar layers lead to emergent and controllable magnetic behavior. However, crystallographic incommensurability and atomic-scale interfacial disorder can severely limit the types of materials amenable to this strategy, as well as the performance of these systems. Here, we demonstrate a method for synthesizing heterostructures comprising magnetic intercalation compounds of transition metal dichalcogenides (TMDs), through directed topotactic reaction of the TMD with a metal oxide. The mechanism of the intercalation reaction enables thermally initiated intercalation of the TMD from lithographically patterned oxide films, giving access to a family of multi-component magnetic architectures through the combination of deterministic van der Waals assembly and directed intercalation chemistry.

Solid heterostructures make it possible to leverage the proximity of disparate electronic phases to engineer exotic physical phenomena[1,2]. Magnetic multilayers are among the most technologically relevant of these systems, with the pioneering discovery of giant magnetoresistance serving as a major milestone in the development of magnetic memory, sensors, and spintronic devices more broadly[3,4]. Beyond elemental multilayers, the assembly of binary or ternary metal compounds into heterostructures vastly expands the range of possible functionality in such systems, as the ground state magnetic/electronic behavior of the constituent layers can be considerably more diverse[5,6].

Equally important is the control over the integrity of interfaces between dissimilar materials. Interfacial atomic disorder–such as that endemic to heterostructures of some oxides and heterojunctions of polar compound semiconductors–can play a defining role in the properties of the architecture[7,8].

Transition metal dichalcogenides (TMDs) intercalated with spin-bearing transition metals are appealing targets for bespoke magnetic heterostructures because they exhibit a wide range of properties that would be attractive in the design and study of magnetic multilayers: hard ferromagnetism in $Fe_{0.25}TaS_2$[9–12], chiral magnetic textures in

[1]Department of Chemistry, University of California, Berkeley, CA, USA. [2]Max-Planck-Institute for Chemical Physics of Solids, Dresden, Germany. [3]Advanced Light Source, Lawrence Berkeley National Laboratory, Berkeley, CA, USA. [4]National Center for Electron Microscopy, Molecular Foundry, Lawrence Berkeley National Laboratory, Berkeley, CA, USA. [5]Research Center for Functional Materials, National Institute for Materials Science, Tsukuba, Japan. [6]International Center for Materials Nanoarchitectonics, National Institute for Materials Science, Tsukuba, Japan. [7]Department of Materials Science and Engineering, Stanford University, Stanford, CA, USA. [8]Chemical Sciences Division, Lawrence Berkeley National Laboratory, Berkeley, CA, USA. [9]Kavli Energy NanoSciences Institute at the University of California Berkeley and the Lawrence Berkeley National Laboratory, Berkeley, CA, USA. ✉e-mail: bediako@berkeley.edu

$Cr_{0.33}TaS_2$ and $Cr_{0.33}NbS_2$ [13–16], electrically driven collinear anti-ferromagnetic switching in $Fe_{0.33}NbS_2$ [17,18], triple-Q antiferromagnetic ordering and topological Hall effect in $Co_{0.33}TaS_2$ and $Co_{0.33}NbS_2$ [19,20], and recently proposed so-called altermagnetic order in $V_{0.33}NbS_2$ [21]. Their diverse properties emerge from the interplay between the host lattice structure, intercalant identity and superlattice symmetry, stoichiometry, and disorder/homogeneity [16,22]. Integrating these versatile crystals into van der Waals (vdW) heterostructures with customizable interfaces may unlock an uncharted frontier of magnetic architectures with designer interfacial interactions and magnetoelectronic properties. However, the absence of effective synthesis methods has impeded the development of 2D heterostructures comprising magnetic intercalated TMDs. Even established techniques for fabricating heterostructures of layered crystals, such as mechanical exfoliation followed by vdW assembly, are not viable for intercalated crystals: the strong interlayer interactions imparted by the intercalants pose significant challenges in isolating thin crystals [23,24], which are integral components of heterostructures. Thus, achieving dimensionality control in heterostructures requires moving beyond the use of flakes exfoliated from bulk intercalated materials. Such exfoliated flakes also inherently exhibit inhomogeneous intercalant distribution on cleaved surfaces [25] and form surface native oxides, hindering the creation of atomically sharp heterointerfaces. Consequently, prior work on vdW assembly of intercalated TMDs has been limited to fabricating magnetic tunnel junctions with incidental oxide formation at poorly defined and controlled heterointerfaces [23,24].

Here, we establish a synthetic framework to create pristine heterostructures comprising magnetic TMD intercalation compounds of $2H$–$TaS_2$ and $2H$–$NbS_2$ containing Fe and Co. These low-dimensional heterostructures exhibit tunable dimensionality, atomically sharp heterointerfaces with modular symmetry and structure, and ultraclean surfaces via deterministic precursor placement. To synthesize these materials, we treat low-dimensional TMD crystals with solutions of zerovalent transition metals. This treatment can lead to TMD intercalation compounds, as demonstrated in previous work where the reaction mechanism remained elusive. We show that this topochemical reaction involves the formation of a metal oxide film derived from the zero-valent metal carbonyl precursor. We then establish that this metal oxide film releases divalent metal ions into the TMD upon annealing. Concomitantly, the oxide-exposed TMD surfaces sacrificially oxidize, thereby supplying the necessary charges to electron-dope the remaining pristine TMD layers to maintain charge balance. Leveraging these mechanistic insights, we develop an approach for patterning metal oxide precursor films onto TMDs that (1) facilitates studies of intercalant diffusion/ordering; (2) produces ultraclean surfaces to allow electronic structure measurements by nano-angle-resolved photoemission spectroscopy (nanoARPES); and (3) enables distinctive magnetic heterostructures of TMD intercalation compounds with atomically clean heterointerfaces. These insights and methodology establish a versatile route to previously inaccessible families of multi-component spintronic architectures through the combination of directed nanoscale solid-state intercalation chemistry with the myriad structural tuning knobs available for vdW heterostructures.

## Results

### Synthesis of few-layer intercalation compounds with solid-state topochemistry

We first illustrate the basic synthetic approach by focusing on a single intercalant species (Fe) and host lattice ($2H$–$TaS_2$), demonstrating how nanoscale solid-state reactions offer a versatile method for synthesizing low-dimensional TMDs intercalated with transition metals. In previous work [12,26] it has been shown that 2D flakes treated with zero-valent metal carbonyls in organic solvents (like $Fe(CO)_5$ in acetone) (Fig. 1a, left) often necessitate vacuum annealing to evince clear signs of

intercalation in their Raman spectra, such as the appearance of new phonon modes associated with the intercalant superlattice and Raman peak shifts indicative of charge transfer to the layered host lattice. Indeed, we find this thermal treatment to be consistently required in the case of $2H$–$TaS_2$ (Supplementary Figs. 1, 3 and 12c) [12]. Here, to first understand the effect of this thermal treatment at an atomic level, we use atomic-resolution high-angle annular dark-field scanning transmission electron microscopy (HAADF-STEM) to examine samples heated to different temperatures. HAADF-STEM imaging was performed on cross-sectional samples made from $2H$–$TaS_2$ flakes on $SiO_2$/Si treated with 10 mM $Fe(CO)_5$/acetone and subjected to thermal annealing at 100 °C (Fig. 1b), 200 °C (Supplementary Fig. 2) or 350 °C (Fig. 1c). These samples, labeled S1–S3, display substantial structural differences: whereas interstitial sites in the vdW interface of S1 and S2 appear largely vacant, S3 displays a high occupancy of intercalants in the pseudo-octahedral sites. To identify these intercalants and compositionally characterize S1–S3, we complement this atomic-resolution imaging with spatially resolved electron energy loss spectroscopy (EELS). This spectroscopic analysis is critical as the intercalants may instead be Ta that has 'self-intercalated' upon heating [27]. EEL spectra of S1–S3 reveal the spectroscopic fingerprint of Fe only for the sample annealed at 350 °C (S3) (Fig. 1d). Congruently, only the Raman spectra of S3 evince peaks shifts and low-frequency modes indicative of intercalation (Supplementary Fig. 3). Taken together, HAADF-STEM, EELS, and Raman data show that substantial intercalation does not occur during treatment in $Fe(CO)_5$ solution alone, even with prolonged (24.5 h) soaking in a 10 mM solution of the zerovalent metal carbonyl. Instead, we observe the formation of a thin film on the surface of the chip (Supplementary Fig. 4) containing Fe, C, and O (Supplementary Note 1, Supplementary Fig. 5). Upon annealing at 350 °C (in a process that necessitates exposing the film for a few minutes to ambient conditions) under high vacuum, $2H$–$TaS_2$ flakes in contact with this film are topochemically converted to $Fe_xTaS_2$ in a solid-state reaction. The chemical details of this topochemical reaction are now discussed.

### Nature of the intercalation reaction with carbonyl-derived films

To probe the redox process accompanying intercalation, we examine the Fe species in the carbonyl-derived films and $2H$–$TaS_2$ using a combination of EELS and X-ray photoelectron spectroscopy (XPS). While $Fe^{3+}$ is the majority species in the films (Supplementary Note 2, Fig. 1e, Supplementary Figs. 6, 7a–d), Fe centers intercalated between $2H$–$TaS_2$ layers exist exclusively as $Fe^{2+}$ (Fig. 1d, e) [28,29], in agreement with prior work [12,30]. We gained additional insight into the intercalation redox chemistry through solid-state reactions between $TaS_2$ and $Fe_2O_3$ powders (Supplementary Note 3, Supplementary Figs. 8–10). These reactions reveal that $Ta^{4+}$ in $TaS_2$ disproportionates to $Ta_2O_5$ and reduced (electron-doped) $TaS_2$ intercalated with $Fe^{2+}$ (i.e., $Fe_xTaS_2$). A possible reaction that describes this process is:

$$(6x + 10)TaS_2 + 5xFe_2O_3 \rightarrow 3xTa_2O_5 + 10Fe_xTaS_2 + 12xS \quad (1)$$

To test this further, we examine the reactions of carbonyl-derived metal oxide films on nano-thick $2H$–$TaS_2$ flakes using high-resolution STEM and EELS measurements (Fig. 1f, Supplementary Fig. 11 and Supplementary Note 4). We find that $2H$–$TaS_2$ layers in direct contact with the films undergo oxidation, while crystalline $Fe_xTaS_2$ layers form underneath (Supplementary Fig. 11), congruent with the proposed disproportionation. These data also suggest that Fe intercalation can take place through the basal plane of $2H$–$TaS_2$, with Fe diffusing both vertically and laterally through $2H$–$TaS_2$ crystals. Moreover, we find that the diffusivity of Fe enables intercalants to move from regions contacting the film to areas partially encapsulated with a chemically inert hexagonal boron nitride (hBN) (Fig. 1f, Supplementary Fig. 12, Supplementary Note 5). Notably, hBN-encapsulated regions do not contain $Ta_2O_5$, demonstrating that pristine $Fe_xTaS_2$ crystals can be

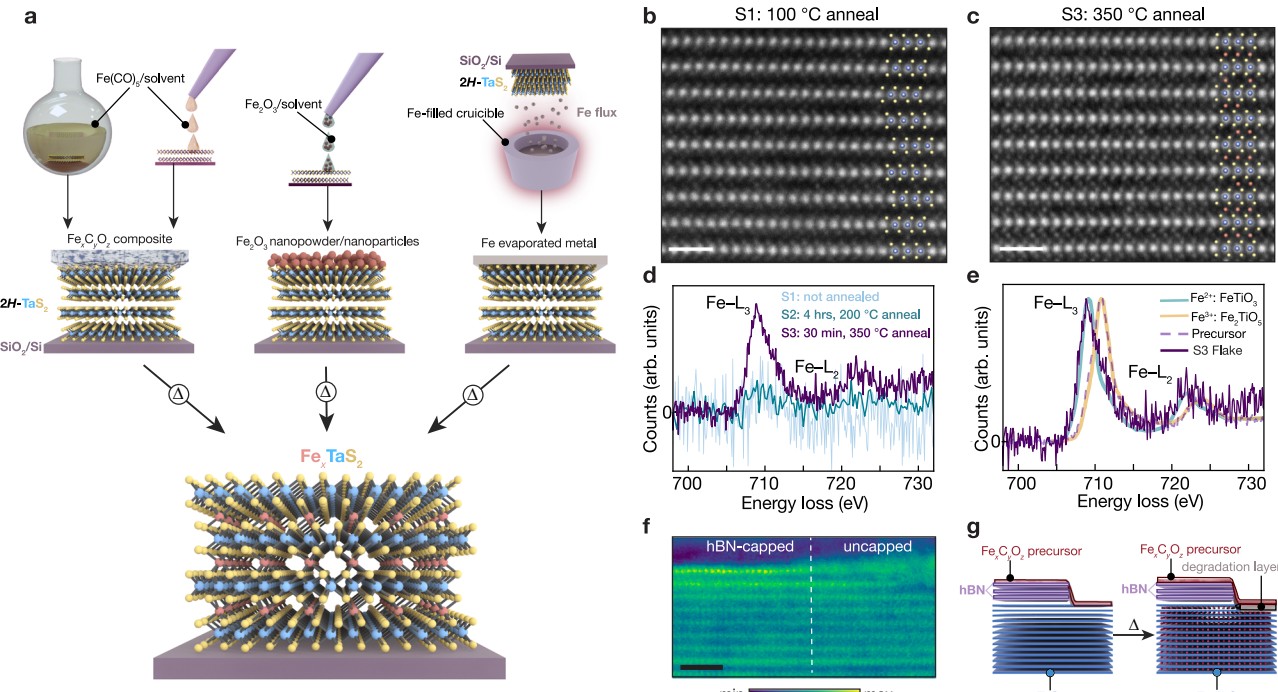

**Fig. 1 | Nanoscale solid-state intercalation of vdW crystals and pristine heterostructures. a** Schematic of depositing Fe precursors onto $2H$−TaS$_2$ crystals on SiO$_2$/Si support followed by vacuum annealing to yield Fe-intercalated $2H$−TaS$_2$ (Fe$_x$TaS$_2$). High-angle annular dark-field scanning transmission electron microscopy (HAADF-STEM) atomic-resolution images along the [10$\bar{1}$0] zone axis of $2H$−TaS$_2$ flakes labeled S1 and S3 immersed in Fe(CO)$_5$/acetone for 24 hours at 48 °C and subsequently vacuum annealed at 100 °C for 12 hours (S1) (**b**) and 350 °C for 30 minutes (S3) (**c**). Micrographs are overlaid with the structure of $2H$−TaS$_2$ (ref. 71). Intercalants are marked in (**c**). The thickness of flakes S1, S2, and S3 are 12 nm, 7.8 nm, and 23 nm, respectively. Scale bars in (**b**, **c**): 1 nm. **d** Cumulative electron energy

loss (EEL) spectra of $2H$−TaS$_2$ flakes labeled S1–S3 treated in Fe(CO)$_5$/acetone and vacuum annealed with distinct thermal conditions. Spectra are normalized by total acquisition time. **e** Cumulative EEL spectra of flake S3 and the composite Fe$_x$C$_y$O$_z$ that formed during the Fe(CO)$_5$/acetone treatment. Experimental spectra, normalized by the L$_3$ peak maxima, are overlaid with literature spectra for (Fe$^{2+}$)TiO$_3$[29] and (Fe$^{3+}$)$_2$TiO$_5$[29]. Data in (**d**) and (**e**) was obtained at ~100 K. **f** High-resolution STEM micrograph of a $2H$−TaS$_2$ flake that was partially capped with hexagonal boron nitride (hBN), treated with Fe(CO)$_5$/isopropanol, and vacuum annealed at 350 °C for 1.5 hours. Scale bar: 2 nm. **g** Schematic of the thermally activated reaction between hBN/$2H$−TaS$_2$ and Fe$_x$C$_y$O$_z$.

---

synthesized by combining vdW heterostructuring (capping with hBN) and this nanoscale solid-state disproportionation chemistry. Thus, although byproducts of disproportionation chemistry are not eliminated, they can be confined to well-defined regions.

To confirm and explore the versatility of this chemistry, we also vacuum annealed mechanically exfoliated $2H$−TaS$_2$ crystals that were coated with a suspension of Fe$_2$O$_3$ nanopowder/nanoparticles or a thin layer of electron-beam evaporated Fe metal that subsequently oxidizes upon air exposure according to XPS in Supplementary Fig. 7e, f (Fig. 1a, middle and right). Both routes also produce Fe$_x$TaS$_2$ (Supplementary Fig. 1a, b). In addition, we find that the carbonyl-derived metal oxide films can simply be prepared by drop-casting or spin-coating solutions of Fe(CO)$_5$ in isopropanol and then exposing these surfaces to air (Fig. 1a, left).

**Intercalation and interlamellar transport from patterned oxide films**

The observation that intercalation proceeds via a solid-state topotactic reaction and not a heterogeneous reaction between solid $2H$−TaS$_2$ and Fe(CO)$_5$ in solution enables the precise patterning of carbonyl-derived oxide films onto 2D flakes (Supplementary Fig. 13). This precursor patterning now allows us to investigate in detail the effect of vacuum annealing conditions on the intercalation, crystallographic ordering of Fe centers in few-layer $2H$−TaS$_2$ flakes, as well as vertical and lateral diffusion of intercalants. We first investigate the vertical diffusion of intercalants by patterning precursors at the center of $2H$−TaS$_2$ flakes, which prevents lateral diffusion from the edges and confines the movement of ions to the out-of-plane direction. The successful

intercalation observed in these samples confirms that vertical diffusion takes place during the intercalation process (Supplementary Fig. 14). Next, we examine the nanoscale structural ordering of Fe centers using temperature-dependent four-dimensional scanning transmission electron microscopy (4D-STEM). In this experiment, a ~6 nm converged electron probe was scanned across a 1.3 $\mu$m × 0.3 $\mu$m window on an 11 nm thick $2H$−TaS$_2$ crystal (through a hole of diameter 2 $\mu$m in the underlying silicon nitride TEM membrane), immediately adjacent to the patterned oxide film. Electron diffraction patterns were obtained at each probe position (Fig. 2a). This experiment was repeated as the temperature of the TEM holder was increased from 14 °C to 350 °C. We find that nanoscale Fe ordering commences at 250 °C, as evidenced by the emergence of faint $\sqrt{3} \times \sqrt{3}$ Fe superlattice diffraction peaks (Fig. 2b). This is consistent with our high-resolution scanning transmission electron microscopy (HRSTEM), Raman spectroscopy and electron energy loss spectroscopy (EELS) experiments: Fe was absent in samples annealed up to 200 °C (Fig. 1b–d and Supplementary Figs. 2, 3).

Further, 4D-STEM analysis revealed that, as expected for any solid-state reaction with a unidirectional reactant source, the amount of ordered Fe increased with higher temperatures and longer annealing times (Supplementary Fig. 15). As these diffraction-based experiments only provide insight into the structurally ordered Fe, 4D-STEM measurements were complemented with STEM-EDS examination of a 22 nm flake annealed ex-situ (Fig. 2c). Compositional analysis of this crystal at three holes in the underlying membrane (centered approximately 1, 5, and 10 $\mu$m away from the oxide film) shows a steady decrease in Fe content ($x$ of 0.30(3), 0.17(2), and 0.04(1), respectively)

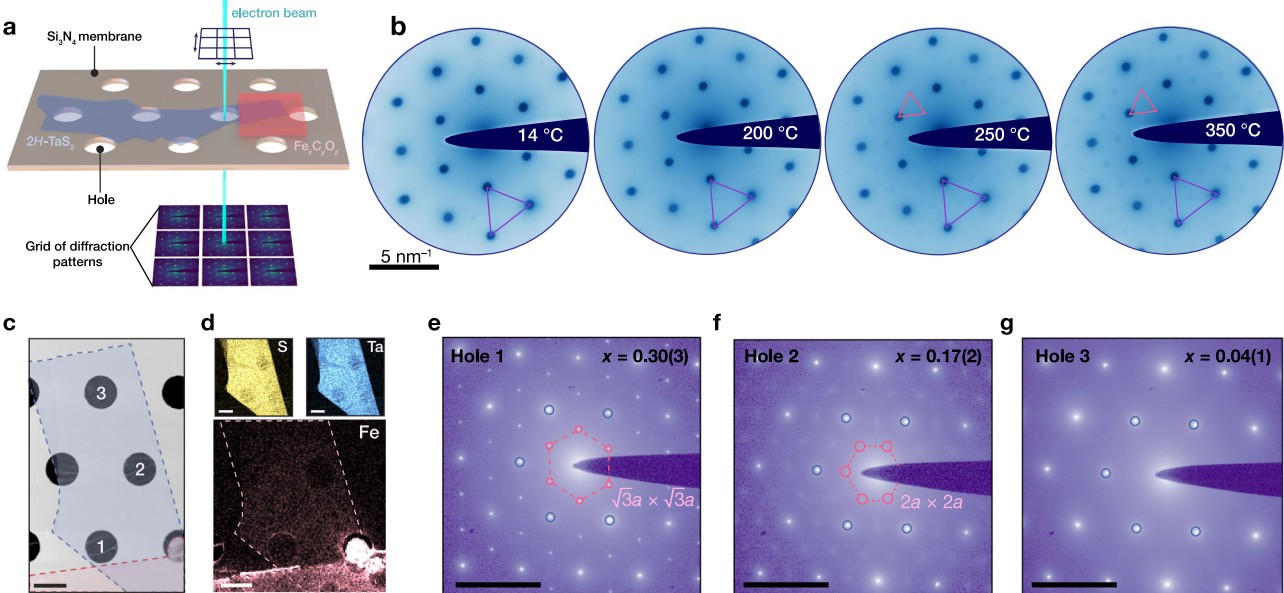

**Fig. 2 | Probing uptake and diffusion of Fe in $2H$–$TaS_2$. a** Schematic illustrating four-dimensional scanning transmission electron microscopy (4D-STEM) of a $2H$–$TaS_2$ flake with a patterned $Fe_xC_yO_z$ precursor. **b** Mean diffraction patterns, displayed on a logarithmic scale, of 4D-STEM datasets obtained during in-situ heating of a $2H$–$TaS_2$ flake with a patterned $Fe_xC_yO_z$ precursor. Diffraction reflections of the $2H$–$TaS_2$ lattice and the Fe superlattice are indicated in violet and red, respectively. The precursor was made by drop-casting a solution of $Fe(CO)_5$/iso-propanol. Data was collected along the $c$-axis of the flake after a 30-minute temperature equilibration period. **c** Plan-view STEM micrograph (∥ $c$-axis) of a $2H$–$TaS_2$ crystal with a patterned $Fe_xC_yO_z$ precursor after ex-situ vacuum annealing for 1 hour at 350 °C. The $Fe_xC_yO_z$ precursor was made by drop-casting a solution of $Fe(CO)_5$/ isopropanol. The $2H$–$TaS_2$ crystal and the $Fe_xC_yO_z$ patterned precursor are false-colored in blue and red, respectively. Number labels for the TEM grid holes are marked in white. Scale bar: 2 $\mu$m (**d**) STEM-EDS maps of sample (**c**). In the Fe EDS map, $2H$–$TaS_2$ is outlined in a dashed line. Scale bars: 2 $\mu$m. Selected area electron diffraction (SAED) patterns of (**c**) obtained for grid hole 1 (**e**), hole 2 (**f**), hole 3 (**g**). The Fe/Ta ratio ($x$) is marked on the SAED patterns. First-order diffraction peaks from the $2H$–$TaS_2$ lattice and the Fe superlattice are highlighted in blue and red, respectively. Scale bars (**e**–**g**): 5 nm$^{-1}$.

with increasing distance from the patterned film (Fig. 2d, Supplementary Fig. 16). Additionally, selected area electron diffraction (SAED) patterns indicate a corresponding change in the Fe superlattice ordering from a well-defined $\sqrt{3} \times \sqrt{3}$ (Fig. 2e) to a diffuse $2 \times 2$ superlattice (Fig. 2f), ultimately disappearing entirely in hole 3 (Fig. 2g). Thus, annealing $2H$–$TaS_2$ samples in contact with a lithographically defined Fe oxide film results in a gradient in Fe content over several microns in distance. Such intercalant gradients can significantly affect magneto-electronic behavior, making their characterization crucial for establishing accurate structure-property correlations. Taken together, our findings reveal that the primary determinants for achieving the desired intercalation products are the annealing temperature, annealing time, positioning of the intercalant oxide film, and the dimensions of $2H$–$TaS_2$ flakes.

## Evolution of electronic band structure of thin $Fe_xTaS_2$ with Fe content

The electronic ramifications of $Fe^{2+}$ intercalation from these oxide films remain unanswered. If our proposed disproportionation chemistry (equation 1) is correct, we would expect electron transfer to the $2H$–$TaS_2$ host. We turn to angle-resolved photoemission spectroscopy (ARPES) measurements with submicron beam diameter to examine the effects of Fe intercalation from $Fe_xC_yO_z$ films on the electronic structure of low-dimensional $2H$–$TaS_2$. A $Fe_xC_yO_z$ film was patterned onto one end of an 11 nm $2H$–$TaS_2$ crystal, which was also partially encapsulated with a monolayer of hBN to prevent degradation of the measured region. This crystal was then annealed in high vacuum for 1 hour at 350 °C. As depicted in Fig. 3a, photoemission spectra as a function of crystal momentum were acquired to measure the electronic band structure of the material over a range of distances away from the patterned $Fe_xC_yO_z$ film with spatial resolution on the order of 1 $\mu$m. We also acquired core-level spectra, confirming the presence of Fe in the

measured area (Supplementary Fig. 17). Representative Fermi surface and energy versus momentum cuts along the $\Gamma$–K direction are presented in Fig. 3b and c, respectively. In measurements of bulk intercalated TMDs, cleaved surfaces can possess two terminations: intercalant termination or TMD termination[25,31,32]. The measured electronic features here are qualitatively consistent with a TMD-terminated surface, which provides evidence for minimal intercalation between the hBN and $H$-$TaS_2$ layers. Interestingly, we also observe that the Fermi surface cuts obtained at different points away from the precursor exhibit distinct intensity distributions (Supplementary Fig. 18), indicating potential variations in electronic reconstruction across the crystal[25].

A quantitative analysis of the nanoARPES datasets collected at various spots across the sample shows the evolution of the band structure with distance from the oxide film. By fitting the momentum distribution curves (MDCs) at the Fermi level ($E_F$) along $\Gamma$–K to Lorentzians (Supplementary Fig. 19), we extracted the Fermi wavevector ($k_F$) to quantify the size of the hole pocket around $\Gamma$. This analysis reveals that the diameter of the central hole pocket increases as a function of distance away from the oxide film, with $k_F$ increasing from 0.35 Å$^{-1}$ to 0.39 Å$^{-1}$ (Fig. 3d). This increase in $k_F$, which is inversely proportional to electron filling, reveals a decrease in electron doping of the $TaS_2$ with distance away from the oxide film (Fig. 3e). This doping trend mirrors the Fe concentration gradient uncovered by Raman mapping of this sample (Supplementary Fig. 20), as well as the STEM EDS and electron diffraction measurements of the sample presented in Fig. 2c–g. As distance from oxide film correlates with the amount of intercalated Fe, these data provide a highly informative picture of the progress of the intercalation reaction itself. As $Fe^{2+}$ is intercalated, the $2H$–$TaS_2$ host lattice is progressively electron-doped. In a simplified picture, this electron transfer to $2H$–$TaS_2$ results in an increased population of the band consisting largely of Ta $d_{z^2}$ orbital

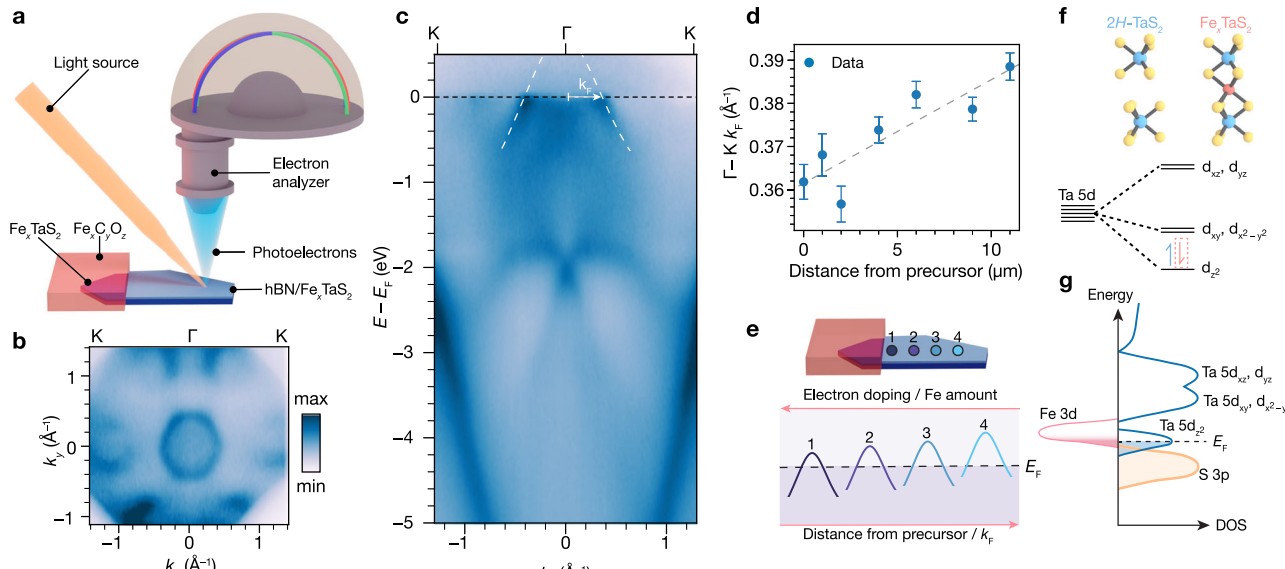

**Fig. 3 | Spatially mapping the band structure of ultra-clean intercalated heterostructures with nanoARPES. a** A schematic depicting the nanoARPES experiment conducted on a $Fe_xTaS_2$ encapsulated with monolayer hBN. The sample was prepared by selectively applying $Fe_xC_yO_z$ onto the area of $2H-TaS_2$ not covered by hBN, followed by vacuum annealing. Normalized ARPES Fermi surface (**b**) and ARPES band dispersion along the $\Gamma-K$ direction (**c**) of a hBN/$Fe_xTaS_2$ heterostructure at a 4 $\mu$m distance from the patterned $Fe_xC_yO_z$ precursor. In (**c**), the Fermi wavevector ($k_F$) is marked, denoting where the band forming the hole pocket around $\Gamma$ intersects the Fermi level ($E_F$). This band is marked by a white dashed line. Data in (**b**, **c**) was obtained at 19 K with h$\nu$ = 118 eV and linear horizontal (LH) polarization. **d** $k_F$ along the $\Gamma-K$ direction extracted from ARPES band-dispersions obtained at different distances from the $Fe_xC_yO_z$ precursor. The $k_F$ values were obtained by fitting the momentum distribution curves (MDCs) at the Fermi level ($E_F$) to Lorentzians. Error bars represent the standard errors for the center positions of Lorentzian peaks corresponding to the main hole pocket around $\Gamma$. The dashed gray line is included as a visual guide. **e** Schematic of the band forming the hole pocket at $\Gamma$ as the distance from the precursor increases from spot 1 to 4. **f** Qualitative d-orbital splitting diagram for the trigonal prismatic Ta center in $Fe_xTaS_2$. A dashed electron in the $d_{z^2}$ orbital denotes additional electron filling upon Fe intercalation, concomitant with charge transfer to $2H-TaS_2$. **g** Qualitative representation of the density of states of Fe-intercalated $2H-TaS_2$.

character, which is at the Fermi level (Fig. 3f, g)[22]. Although the experimental band structures of TMDs involve more complexity than this basic explanation[25,33], increasing $Fe^{2+}$ content leads to increased electron doping that in turn reduces the size of the central hole pocket of this Ta $d_{z^2}$ band (Fig. 3e). Therefore, the data of Figs. 2 and 3 show how the $Fe^{2+}$ content has structural and electronic implications, and the interplay between these factors is crucial for explaining the emergent magnetism in these materials[22].

## Intercalation compound heterostructures

Having unveiled this topotactic chemistry for intercalating TMDs with open-shell metal ions, we exploit this method to craft heterostructures of magnetic intercalation compounds. We demonstrate two paradigms for this combination of topotactic chemistry and vdW heterostructure assembly. In the first case, we intercalate a TMD heterostructure ($2H-TaS_2$/$2H-NbS_2$) with a single intercalant element (Fe), and in the second case, we create heterostructures using two different intercalants (Fe and Co) in a stack of the same TMD, $2H-TaS_2$.

$Fe_xTaS_2$ is ferromagnetic (FM) in the bulk[9–11] as well as in the few-layer limit[12]. In contrast, $Fe_yNbS_2$ is known to be antiferromagnetic (AFM) in the bulk[17,18]. Consequently, the synthesis of $Fe_xTaS_2$/$Fe_yNbS_2$ heterostructures with clean interfaces would enable the construction of FM/AFM bilayers that may display technologically relevant magneto-electronic properties as a result of interfacial coupling between the dissimilar magnetic states[34–36]. To synthesize clean $Fe_xTaS_2$/$Fe_yNbS_2$, interfaces, we prepare $2H-TaS_2$/$2H-NbS_2$ heterostructures using vdW assembly, deposit a $Fe_xC_yO_z$ film from Fe(CO)$_5$/toluene solution, and anneal the heterostructure at 350 °C for 1.5 hours (Fig. 4a). STEM-EDS mapping confirms the presence of Fe across the resulting heterostructure (Fig. 4b) and estimates the Fe content to be 0.42 and 0.32 per $2H-NbS_2$ and $2H-TaS_2$, respectively. The presence of low frequency

Raman modes at 139 cm$^{-1}$ and 181 cm$^{-1}$ (Fig. 4c, Supplementary Fig. 24, Supplementary Fig. 25) are consistent with $\sqrt{3} \times \sqrt{3}$ Fe superlattices in $Fe_xTaS_2$ and $Fe_yNbS_2$, respectively[12,37,38]. HRSTEM imaging of this $Fe_{0.32}NbS_2$/$Fe_{0.30}TaS_2$ assembly reveals an atomically flat and sharp heterointerface between the dissimilar 2D crystals (Fig. 4d,e) with no amorphous oxide tunneling barrier (typically observed in heterostructures made from cleaved bulk crystals)[23,24]. Moreover, atomic-resolution HRSTEM confirms that Fe ions decorate the interstitial between all vdW layers, including the 3.5-degree twisted heterointerface of Fe-intercalated $2H-TaS_2$ and $2H-NbS_2$ crystals (Fig. 4e).

To create co-intercalated heterostructures, we pattern carbonyl-derived oxide films onto TMD stacks. This multi-precursor technique was applied to two stacked $2H-TaS_2$ flakes, with oxide precursors $Fe_xC_yO_z$ and $Co_xC_yO_z$ (Supplementary Fig. 21) films patterned on the top and bottom $2H-TaS_2$ crystals, respectively (Fig. 4f, g). After thermal annealing at 350 °C, we examine the overlapping region of the $2H-TaS_2$ flakes using STEM-EDS (Fig. 4h–j, Supplementary Fig. 22). These measurements show that the top flake is principally Fe-rich and the bottom flake is Co-rich. Notably, although Fe and Co are present in both flakes, each flake contained a substantially higher concentration of the metal patterned directly onto it, implying that the lateral diffusion is considerably more facile than the vertical diffusion. This observed intercalant segregation distinguishes these heterostructures from bulk crystals grown with solid-state methods and singular flakes intercalated with two precursors (Supplementary Fig. 23), where multiple metal intercalants disperse uniformly throughout the host lattice[39]. Atomic-resolution HAADF-STEM imaging of our heterostructures provides an explanation for this segregation (Fig. 4k), showing that the interface between the $2H-TaS_2$ crystals is atomically sharp and, importantly, azimuthally misaligned (twisted). Twisting creates an in-plane moiré superlattice[40,41] and necessarily reduces the

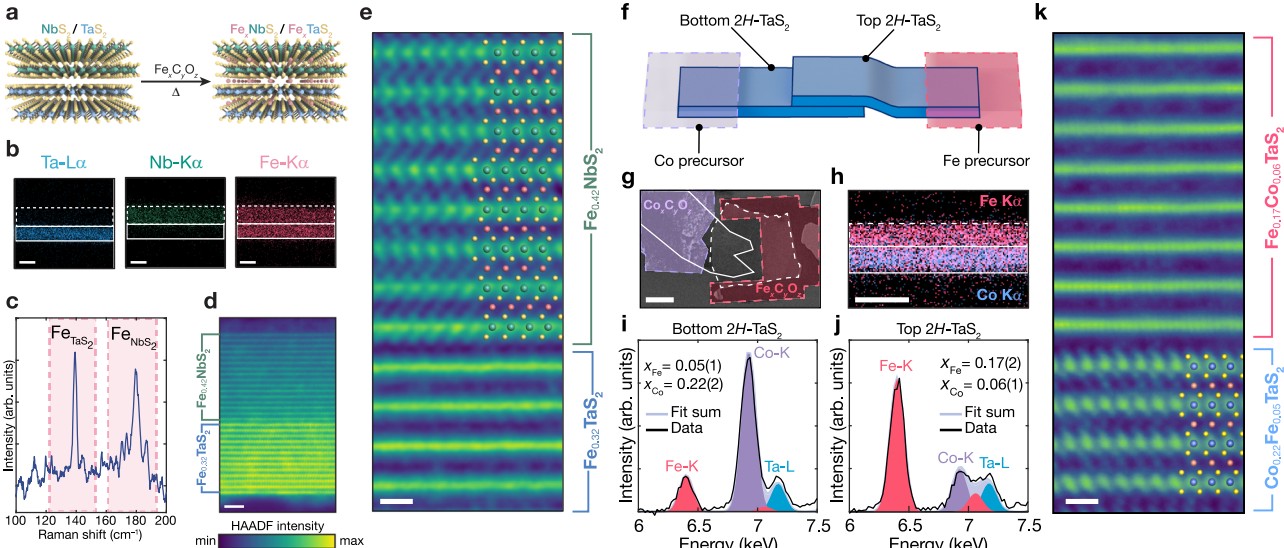

**Fig. 4 | Synthesis and characterization of intercalated heterostructures.**
**a** Schematic illustrating the synthesis of Fe-intercalated heterostructure of $2H$–NbS$_2$/$2H$–TaS$_2$. **b** STEM energy-dispersive X-ray spectroscopy (EDS) maps perpendicular to the $c$-axis of the heterostructure synthesized as depicted in (**a**). The dashed and solid white lines mark the edges of $2H$–NbS$_2$ and $2H$–TaS$_2$, respectively. Scale bars: 10 nm. **c** Ultra-low frequency (ULF) Raman spectra of the heterostructure from (**b**). **d** High-resolution STEM image of the Fe$_{0.42}$NbS$_2$/Fe$_{0.32}$TaS$_2$ heterostructure from (**b**) along the [10$\bar{1}$0] zone axis of the TaS$_2$ flake. The azimuthal misalignment (twist) angle is 3.5°, determined using Kikuchi bands. Scale bar: 2 nm. **e** Atomic-resolution HAADF-STEM image of the heterostructure from (**b**) obtained along the [10$\bar{1}$0] zone axis of the NbS$_2$ flake. Scale bar: 5 Å. Crystal structure of Fe$_{1/3}$NbS$_2$ from ref. 72 is overlaid with the $2H$–NbS$_2$ flake. **f** Illustration of a heterostructure comprising two $2H$–TaS$_2$ flakes with the top and bottom one

partially covered with the Fe$_x$C$_y$O$_z$ and Co$_x$C$_y$O$_z$ precursors, respectively. **g** Plan-view (∥ $c$-axis) scanning electron microscope (SEM) image of a heterostructure made according to the synthetic design in (**f**). The heterostructure with the patterned precursors was annealed for 1 hour at 350 °C. In the SEM image, $2H$–TaS$_2$ flakes are outlined in white, while the Co and Fe precursors are false-colored in violet and red, respectively. Scale bar: 5 µm. **h** STEM-EDS map of a cross-sectional TEM sample made from heterostructure (**g**). Scale bar: 20 nm. Cumulative STEM-EDS spectra of the bottom (**i**) and top (**j**) $2H$–TaS$_2$ flakes from (**g**). In (**i**, **j**), marked $x_M$, where $M$ = (Fe, Co), denotes the stoichiometric ratio between $M$ and Ta. **k** Atomic-resolution HAADF-STEM image of the heterostructure from (**g**) obtained along the [10$\bar{1}$0] zone axis of the bottom $2H$–TaS$_2$ flake. The azimuthal angle (twist) between the $2H$–TaS$_2$ flakes is 20°. Crystal structure of Fe$_{1/3}$TaS$_2$ from ref. 73 is overlaid with the bottom TaS$_2$ flake. Scale bar: 5 Å.

availability of optimal pseudo-octahedral intercalant sites. These results raise the possibility that controlling these azimuthal angles may be used to shape the gradient of co-intercalants and the resulting magnetism in tailored mixed heterostructures.

To demonstrate the magneto-electronic functionality facilitated by this approach, we conducted electronic transport assessments of the Fe$_{0.42}$NbS$_2$/Fe$_{0.32}$TaS$_2$ heterostructure that was presented in Fig. 4b–e. Figure 5a shows the fabricated mesoscopic device (labeled D1), comprising three distinct measurement regions, including Fe-intercalated $2H$–TaS$_2$ (region R1), Fe-intercalated $2H$–NbS$_2$ (region R3), and their composite heterostructure (region R2) (Fig. 5a). In regions R1–R3, the temperature-dependent longitudinal resistance reveals an inflection near 30 K (Supplementary Fig. 26), suggesting a phase transition upon cooling. The magnetic ordering temperatures of R1–R3 are 30 K, 34 K, and 32 K, respectively, as determined from the inflection in $R_{xx}$ observed upon zero-field cooling[9,17,42] (Supplementary Fig. 26). Additionally, upon cooling in a 12 T magnetic field, these inflections broaden and shift to higher temperatures for R1–R2 and lower temperatures for R3 (Supplementary Fig. 26), implying the magnetic origin of the phase transition consistent with ferromagnetism[10,43] in R1–R2 and antiferromagnetism in R3[44,45].

Further understanding of the long-range magnetic ordering in R1–R3 was gained through measurements of the magnetoresistance (MR) and field-dependent anomalous Hall resistance ($R_{AHE}$) (Fig. 5b, c, Supplementary Figs. 27, 28), which reflect the interactions between the itinerant hole carriers (Supplementary Fig. 29) and the magnetic moments. Interestingly, hysteretic MR and remnant $R_{AHE}$ remain observable up to 40 K for R1–R2 (Fig. 5c, Supplementary Figs. 27,28) consistent with the onset of broken time reversal symmetry at temperatures higher than those estimated from the inflection in resistance upon zero-field cooling. It is noteworthy that all regions exhibit a

strong response to an out-of-plane magnetic field ($H\parallel c$) while showing negligible response to an in-plane field ($H\parallel ab$) (Supplementary Fig. 31). This confirms their strong magnetocrystalline anisotropy, a characteristic imperative for stabilizing magnetism in the ultrathin limit.

First, we turn to the detailed analysis of the magnetotransport of R1 and R3, which consist of a singular host lattice. Region R1 (Fe$_{0.32(1)}$TaS$_2$) manifests a hysteretic negative magnetoresistance with a bow-tie-shaped profile, characteristic of ferromagnets[10,12] (Fig. 5b, c and Supplementary Fig. 28a). Moreover, the presence of hysteresis in its $R_{AHE}$ signal, which is proportional to magnetization, further supports the ferromagnetic characteristics of R1[10,12] (Fig. 5b, c and Supplementary Fig. 28a). Conversely, the magnetoresistance (MR) behavior of R3 (Fe$_{0.32(3)}$NbS$_2$) mirrors that of metallic antiferromagnets, demonstrating positive MR below the Néel temperature and transitioning to negative MR above 40 K[44,46,47] (Fig. 5b, c and Supplementary Fig. 28c). However, contrary to the expected behavior for antiferromagnets, R3 also exhibits hysteresis in its MR profile below the ordering temperature, suggesting the presence of uncompensated magnetic moments. Similar hysteretic behavior, albeit only in the vicinity of the Néel temperature, has been observed in bulk flakes of Fe$_{0.33}$NbS$_2$[46] and is attributed to a minority uncompensated magnetic phase. Given that our samples deviate from stoichiometric Fe$_{0.33}$NbS$_2$, we propose that these uncompensated moments arise from intercalant disorder, which is known to introduce a spin glass phase in bulk crystals of Fe$_x$NbS$_2$[18,48]. The magnetic relaxation behavior of R3 supports the presence of glassy magnetic behavior (Supplementary Note 6, Supplementary Figs. 32–34). We also note that the uncompensated phase could be stabilized through field-driven metamagnetic transitions (Supplementary Note 7, Supplementary Fig. 36), which are observed in bulk crystals of Fe$_x$TiS$_2$[49–52]. While further investigation is needed to fully elucidate the origin and nature of the uncompensated

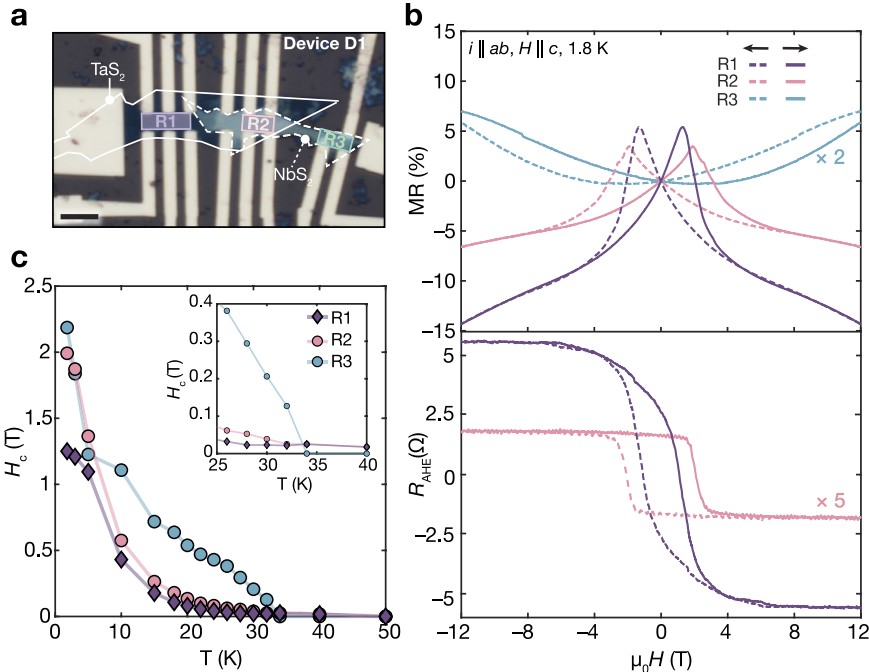

**Fig. 5 | Magnetotransport of iron-intercalated 2*H*–NbS₂/2*H*–TaS₂ hetero-structures. a** Optical micrograph of the measured mesoscopic device. Scale bar: 5 $\mu$m. Three distinct measured regions are false-colored and labeled: R1–Fe$_{0.32(1)}$TaS₂, R2–Fe$_{0.42(4)}$NbS₂ / Fe$_{0.31(2)}$TaS₂ and R3–Fe$_{0.32(3)}$NbS₂. **b** Field-dependent magneto-resistance (MR) and anomalous Hall resistance ($R_{AHE}$) recorded in regions R1–R3. **c** Temperature-dependent coercive field ($H_c$) for R1–R3. For R1–R2, the average of

$H_c$ is obtained from both MR and $R_{AHE}$ measurements. Here, $H_c$ refers to the field where MR reaches its maximum while $R_{AHE}$ equals zero. Conversely, for R3, $H_c$ is determined exclusively from MR data, defining it as the field where MR exhibits its minimum (upturn). The data presented in (**b**, **c**) was obtained after the initial thermal and field cycling of the device.

phase in R3, our data suggests that it coexists with AFM order and engenders the observed hysteretic behavior.

Having established R1 as ferromagnetic and R3 as anti-ferromagnetic with a coexisting uncompensated phase, the transport behavior of the heterostructure (R2) was examined (Fig. 5b, c and Supplementary Fig. 28b). The magnetotransport characteristics of R2 are dominated by the Fe$_{0.31(2)}$TaS₂ segment of the heterostructure, aligning with the ferromagnetic behavior observed in R1 (Fig. 5b, c and Supplementary Fig. 28a, b). However, following the initial thermal and field cycling, the coercive fields ($H_c$) of R2 consistently surpass those of R1 below the Néel transition of R3 (Fig. 5c, Supplementary Fig. 37). This suggests that out-of-plane interactions across the ferromagnetic-antiferromagnetic heterostructure may be responsible for the high coercivity of R2. Consequently, the hetero-structure behaves akin to an exchange spring magnet, a composite of dissimilar magnets whose interfacial interactions increase the coercivity[53]. These results underscore the potential of intercalated heterostructures, which are enabled by combining nanoscale topo-chemical reactions and vdW heterostructure assembly, as a versatile platform for designing and interrogating low-dimensional magnetic phenomena.

## Discussion

This study demonstrates the successful synthesis of TMD hetero-structures intercalated with open-shell transition metal ions by implementing vdW assembly and nanoscale solid-state topochemistry of the TMD with a transition metal oxide film. The intercalation process is comprehensively investigated for Fe$_x$TaS₂, using both bulk and atomic-scale structural and spectroscopic probes. We uncover that intercalation is thermally activated and accompanied by the dis-proportionation of oxide-exposed 2*H*–TaS₂ layers. We observe that the intercalants diffuse from the region in contact with the oxide film, allowing for the fabrication of pristine few-layer magnetic intercalation

compounds under hBN-encapsulated flakes. Leveraging these insights, we use patterning of metal carbonyl-derived oxide films to direct metal ions into well-defined vdW heterostructures, expanding the range of accessible low-dimensional magnetic intercalation compounds as well as producing magnetic multilayer systems with exceptionally clean surfaces. Representative measurements of such het-erostructures show transport behaviors signifying strong interfacial magnetic exchange effects that are only possible with atomically clean interfaces.

We envision the emergent physics of these heterostructures may be controlled by altering the thickness of constituent flakes and the intercalants' identity, stoichiometry, and homogeneity. Additionally, we expect that the synthesis of heterostructures with atomically sharp moiré heterointerfaces will offer exciting opportunities to both con-trol the intercalation chemistry and also to fundamentally engineer complex magnetic behaviors. This customizability of vdW hetero-epitaxy combined with nanoscale intercalation chemistry unlocks versatile opportunities to fine-tune the spin degree of freedom in materials. As follows, intercalated heterostructures are compelling platforms for multi-component spintronic device architectures, including spin-orbit torque devices, magnetic tunnel junctions, and memristors[54].

## Methods

### Synthesis of 2*H*–TaS₂ bulk crystals

Single crystals of 2*H*–TaS₂ were grown with chemical vapor transport (CVT) using powders of elemental Ta (−100 mesh, 99.98%, Nb 50 ppm, Alfa Aesar), S (99.999%, Acros Organics), and I₂ (99.999%, Spectrum Chemicals), which were used as received. Elemental Ta (176.3 mg, 1.00 equiv.) and S (62.7 mg, 2.00 equiv.) were sealed in a fused quartz ampule (14 mm inner diameter, 1 mm wall thickness, 15 cm long) under vacuum ( ~ 1 × 10⁻⁵ Torr), along with 152.1 mg I₂ (5 mg/cm³). The ampule was placed in an MTI OTF-1200X-II two-zone tube furnace with the hot

zone maintained at 1000 °C and the cold (growth) zone maintained at 850 °C for 6 days, before cooling to room temperature at approximately 60 °C/hour.

## Mechanical exfoliation of 2H−TaS₂ and 2H−NbS₂

Exfoliation of $2H$−$TaS_2$ (synthesized in this study or obtained from HQ Graphene) and $2H$−$NbS_2$ (HQ graphene) was performed in an Ar-filled glove box with Magic Scotch adhesive tape. Crystals were exfoliated onto 90 nm $SiO_2$/Si wafers (Nova Electronic Materials), which had been cleaned in an oxygen plasma cleaner for 2 minutes and then heated to 200 °C on the glove box hotplate.

## Mechanical exfoliation of hexagonal boron nitride (hBN)

Exfoliation of hBN (used as received from T. Taniguchi and K. Watanabe) was conducted using Magic Scotch tape in ambient conditions onto a 90 nm $SiO_2$/Si wafer (Nova Electronic Materials), which had been subjected to a 90-minute ozone cleaner treatment at 150 °C.

## Determining the thickness of flakes

The thickness of flakes used for cross-sectional imaging was obtained using HRSTEM after cross-sectioning. For other flakes, atomic force microscopy (Park Systems NX10) was used to determine the sample thickness.

## Preparation of $M_xC_yO_z$ ($M$ = Fe, Co) films

To prepare $M_xC_yO_z$ ($M$ = Fe, Co), the following chemicals were used as received: iron(0) pentacarbonyl (>99.99 %, Sigma-Aldrich), dicobalt octacarbonyl (moistened with hexane (1-10%), ≥90 % (Co)) acetone (≥99.5 %, Fisher Scientific), and isopropanol (IPA) (≥99.5 %, Sigma-Aldrich). Acetonitrile (≥99.5 %, Fisher Scientific) and toluene (≥99.5 %, Fisher Scientific) were obtained from a Pure Process Technology solvent purification system, and acetone (Sigma-Aldrich) was degassed by the freeze-pump-thaw technique and then dried over $CaCl_2$ (Acros Organics). All glassware was oven-dried at 150 °C for a minimum of 2 hours before use. Two methodologies were used for preparing $M_xC_yO_z$ ($M$ = Fe, Co): (1) prolonged soaking in a metal carbonyl solution and (2) drop-casting a metal carbonyl solution.

In the soaking method, a 10 mM solution of $Fe(CO)_5$ or $Co_2(CO)_8$ was prepared in toluene, acetonitrile, or acetone. Experiments in toluene were performed in an Ar-filled glove box, while experiments in acetonitrile and acetone were conducted on a Schenk line. Mechanically exfoliated $2H$−$TaS_2$ on a supporting substrate was immersed in the metal carbonyl solution for 14−48 hours, followed by three rinses with fresh solvent and then drying using an inert gas.

For the drop-casting approach, a solution of $Fe(CO)_5$ (380 mM) or $Co_2(CO)_8$ (2.5 mM) in IPA was prepared on a Schlenk line. To dissolve $Co_2(CO)_8$, the solution was stirred at 70 °C under $N_2$ for approximately 14 hours. Subsequently, the metal carbonyl solution was drop-cast onto mechanically exfoliated $2H$−$TaS_2$ while heating the sample to 80 °C in air.

## Treatment of 2H−TaS₂ with evaporated Fe

In a typical procedure, 10 nm of Fe was evaporated at 2 Å/second onto mechanically exfoliated $2H$−$TaS_2$ flakes in an e-beam evaporator under high-vacuum conditions ( ~ $10^{-6}$ Torr). Following the evaporation, the chips were moved in the air to our home laboratory and stored in an Ar-filled glove box.

## Treatment of 2H−TaS₂ with Fe₂O₃ nanoparticles/nanopowder

A dispersion of $Fe_2O_3$ nanopowder (Sigma Aldrich, <50 nm particle size (BET)) in IPA or a dispersion of $Fe_2O_3$ nanoparticles in ethanol (Sigma Aldrich, ≤110 nm particle size, 15 wt. % in ethanol, ligands: [2-(2-Methoxyethoxy)ethoxy]acetic acid) was drop cast onto mechanically exfoliated $2H$−$TaS_2$ flakes.

## Annealing samples after precursor treatment

Samples in contact with solid precursors were annealed in high-vacuum (~$10^{-7}$ Torr) by rapidly warming to 120 °C at 60 °C/min with a 5-minute hold, followed by heating at 11.5 °C/min to 350 °C, and hold for 30 minutes to 1.5 hours at 350 °C, before cooling to room temperature at 13.5 °C/min.

## Patterning of $M_xC_yO_z$ precursors

E-beam lithography (PMMA 950 A6 resist) was used to pattern $M_xC_yO_z$ films onto TMD heterostructures (Supplementary Fig. 13). Subsequently, a solution of $Fe(CO)_5$ (380 mM) or $Co_2(CO)_8$ (2.5 mM) in IPA was drop-cast onto the lithographically-patterned chip. The PMMA was then lifted off by soaking the sample in acetone, followed by cleaning the sample in IPA and drying it with an $N_2$ jet. Samples patterned with both $Fe_xC_yO_z$ and $Co_xC_yO_z$ films were prepared by following this procedure twice to separately pattern the dissimilar metal oxide films.

## Synthesis of $Fe_xC_yO_z$ bulk powders

Under an Ar atmosphere, iron(0) pentacarbonyl (5 mL, 38 mmol) was added to a glass vial, and mixed with 15 mL of toluene or 2-propanol on a Schlenk line. Under an $N_2$ atmosphere, the solution was allowed to dry at room temperature for approximately 2-3 days.

## Solid-state reactions of polycrystalline 2H−TaS₂ and Fe₂O₃

First, polycrystalline $TaS_2$ was prepared by grinding a stoichiometric mixture of Ta (~100 mesh, 99.98%, Nb 50 ppm, Alfa Aesar) and S (99.999%, Acros Organics) powders in a mortar and pestle for 10 minutes. This mixture was subsequently pressed into a pellet and sealed in a glass ampoule (14 mm inner diameter, 1 mm wall thickness, 10 cm long) under vacuum (~ $1 \times 10^{-5}$ Torr). The sealed ampoule was heated in a muffle furnace at 900 °C for 7 days with 1 °C/minute cooling and warming rates. The resulting polycrystalline powder was analyzed with powder x-ray diffraction, PXRD (Supplementary Fig. 9), and then mixed with $Fe_2O_3$ nanopowder (Sigma Aldrich, <50 nm particle size (BET)) in 4:1 and 7:1 molar ratios. These powder mixtures were then sealed in quartz ampoules and heated as described above for polycrystalline $TaS_2$. The products were analyzed using Raman spectroscopy and PXRD (Supplementary Figs. 8, 10).

## Sample preparation for transmission electron microscopy (TEM)

Samples for imaging along the $c$-axis were prepared in an Ar-filled glovebox using the dry transfer method[12,55]. Flakes were transferred onto a 200 nm silicon nitride holey TEM grid (Norcada), treated with an $O_2$ plasma for 5 minutes immediately before stacking.

Cross-sections of each flake for imaging along the crystallographic $ab$-plane were prepared by standard focused ion beam (FIB) lift-out procedure using Thermo Fisher Scientific Helios G4 and Scios 2 FIB-SEM systems[12,55]. Prior to imaging at the National Center for Electron Microscopy (NCEM), samples underwent additional cleaning using the Fischiotone Ar nanomill set at milling angles of +15 deg and −10 degrees. Plasma cleaning of the milling holder was carried out beforehand to mitigate potential carbon contamination. Before atomic-resolution STEM imaging, TEM lamellae were annealed in high-vacuum ($10^{-7}$ Torr) at 100−120 °C for 12 hours to decrease carbon contamination.

## Transmission electron microscopy (TEM)

Atomic-resolution high-angle annular dark-field STEM imaging of cross-sections presented in Fig. 1b,c and Supplementary Fig. 2 was performed on a Thermo Fisher Spectra 300 X-CFEG operating at 120 kV with 24 mrad probe-forming convergence angle, using inner and outer collection angles of 68 and 200 mrad, respectively. The remaining HAADF-STEM imaging was conducted on TEAM I, a modified FEI Titan 80-300 microscope, operating at 80 kV with a beam current of 70 pA. The convergence angle was set to 30 mrad for data

shown in Figs. 1f, 4k, and Supplementary Fig. 12b. A convergence angle of 17 mrad was used to collect data in Fig. 4d, e. For atomic resolution imaging, stacks of images with a high frame rate (>1 fps) were rigidly registered and summed[56] to increase the signal-to-noise ratio without introducing artifacts from sample drift or scan noise.

Core-loss EELS measurements were performed on $2H$–$TaS_2$ flakes after treatment with $Fe(CO)_5$/acetone and subsequent annealing. Data were collected at cryogenic temperatures to reduce possible effects of radiation damage in the sample during long acquisitions using a Gatan 636 side-entry liquid nitrogen holder in an FEI Titan Themis CryoS/TEM operating at 120 kV equipped with a 965 GIF Quantum ER and a Gatan K2 Summit direct electron detector operating in electron counting mode. EEL spectrum images extending from the substrate into the protective surface layers and lateral distances spanning tens of nanometers were acquired throughout several regions of each sample. Spectra were summed over selected areas from each map to extract different regions for analysis—e.g., within the TMD flake or the precursor film—using Nion Swift image processing software[57]. These spectra were then combined across data sets from a given sample to produce high signal-to-noise ratio summed spectra. The energy alignment of EEL spectra was corrected using the well-defined $Ca^{2+}$ edge as a reference. The $Ca^{2+}$ signal was evident in the $Fe_xC_yO_z$ film because the acetone used was dried over $CaCl_2$.

Selected area electron diffraction (SAED) and STEM-EDS data were collected in FEI TitanX operating at 80 kV. SAED patterns were obtained with a 40 $\mu$m diameter aperture, which defined a selected diameter of ~ 720 nm. STEM-EDS data was collected with a probe convergence angle of 10 mrad and a beam current of 150 pA–1 nA. The total acquisition time was 3–30 minutes, depending on the signal-to-noise ratio. Data was analyzed in the Bruker Esprit 1.9 software, and series fit deconvolution was implemented to resolve overlapping peaks. We note that the sulfur content of cross-sectional TMD samples varied significantly, ranging from 1 to 1.9 stoichiometric units, with thicker lamella regions having a higher sulfur content. In contrast, the sulfur content of plan-view samples is more consistent, ranging from 1.75 to 1.95 stoichiometric units. Thus, we hypothesize that the sulfur content may be significantly impacted by the FIB preparation process and/or the bakeout steps before the imaging.

Four-dimensional scanning transmission electron microscopy (4D-STEM) was performed in an FEI TitanX (80 keV, 0.55 mrad indicated convergence semi-angle) using a Gatan 652 Heating holder for in-situ heating experiments. The collected data was analyzed using the py4DSTEM Python package[58]. First, we used peak detection to locate the $2H$–$TaS_2$ Bragg peaks, used to subsequently construct the reciprocal vectors of the $2H$–$TaS_2$ host lattice. Next, $\sqrt{3} \times \sqrt{3}$ Fe superlattice reciprocal vectors were calculated from the symmetry relations to the $2H$–$TaS_2$ reciprocal vectors. Note, the $\sqrt{3} \times \sqrt{3}$ Fe superlattice was the only detectable Fe superlattice within our 4D-STEM datasets. Subsequently, we mathematically constructed virtual apertures that masked the entire diffraction space except for the Fe superlattice peak regions (Supplementary Fig. 15a). The constructed virtual apertures were applied to the 4D-STEM diffraction data to integrate the Fe superlattice intensities at each probe position. A background (bg) virtual aperture was also defined to estimate the diffuse scattering background, subtracted from the integrated Fe superlattice intensities at each probe position (Supplementary Fig. 15a).

## X-ray photoelectron spectroscopy (XPS)
All samples, transported in Ar-filled bags, were mounted onto stainless-steel stubs using double-sided tape under ambient conditions, which they were exposed to less than 5 minutes before loading into the XPS chamber. Measurements were performed using a Thermo Scientific K-Alpha+ (Al K$\alpha$ radiation, h$\nu$ = 1486.6 eV) (Thermo Fisher Scientific Inc) equipped with an electron flood gun for charge neutralization. The AvantageTM package was used for data acquisition. Spectra were

collected with a fixed-analyzer transmission mode with an X-ray beam size of 400 $\mu$m. Survey scans were collected with a pass energy of 200 eV, while high-resolution scans of individual elements were collected with a pass energy of 20–50 eV and a step size of 0.1 eV. Auto-Z (i.e., automated height adjustment to the highest intensity) was performed before each measurement to find the focal point of the analyzer. We collected 30 sweeps for measuring Fe and 40 sweeps for measuring Co to obtain the optimal signal-to-noise ratio. Data were analyzed using Thermo Scientific Avantage Data System software (version 5.9914), where a smart background was applied before peak deconvolution and integration. All spectra were charge corrected to the C 1s peak at 284.8 eV. The Fe 2p and Co 2p peak positions were assigned using established literature[59–67].

## Powder X-ray diffraction (PXRD)
PXRD data was collected in ambient conditions on a Bruker D8 Advance diffractometer using Cu K$\alpha$ ($\lambda$ = 1.5406 Å) radiation.

## Fourier transform infrared spectroscopy (FTIR)
Fourier transform infrared spectroscopy measurements were carried out on a Thermo Scientific Nicolet iS20 spectrometer utilizing KBr pellets. The preparation of samples for infrared spectroscopy was carried out in an Ar-filled glovebox.

## Scanning electron microscopy (SEM) and energy-dispersive X-ray spectroscopy (EDS)
Compositional analysis of bulk crystals with SEM-EDS was performed using a Thermo Scientific Scios 2 FIB-SEM system with an accelerating voltage of 20–30 kV and an average beam current of 1.6 nA.

## Confocal Raman spectroscopy
Ultra-low frequency (ULF) Raman spectroscopy (Horiba Multiline LabRam Evolution) was conducted using a 633 nm laser excitation at a power of 50–100 $\mu$W with 10–60 second acquisition times and 3–6 accumulations.

## Nanofabrication of devices
To create electrical leads, samples were spin-coated with PMMA 950 A6 (Kayaku Advanced Materials) at 4000 rpm and cured on a hot plate in ambient conditions at 180 °C for 3 minutes. Next, electron beam lithography (Crestec CABL-UH Series Electron Beam Lithography System) was used to define the electrical contacts, which were treated with reactive ion etching (RIE) with a mixture of 70 sccm $CHF_3$ and 10 sccm $O_2$ (SEMI) for 1 minute to expose a fresh surface of the sample. Immediately after etching, Cr/Au (2 nm/100 nm) was thermally evaporated onto the sample and lifted off in acetone overnight.

## Electron transport measurements
Four-probe transport measurements were conducted with conventional lock-in methods. In summary, a 5 $\mu$A alternating current (17.777 Hz) was introduced between the source and drain terminals while systematically varying the temperature using the PPMS DynaCool system. Simultaneously, the longitudinal ($V_{xx}$) and transverse voltage ($V_{xy}$) were recorded with the SR830 lock-in amplifier. Phase differences remained below 5, and resistances were computed following Ohm's law. Unless otherwise indicated, the field-dependent $R_{xx}$ and $R_{xy}$ data were symmetrized (anti-symmetrized) in accordance with established methods[12,68]:

$$R_{xx,symm}^{Forward}(\mu_0 H) = \frac{1}{2}\left[R_{xx,raw}^{Forward}(\mu_0 H) + R_{xx,raw}^{Reverse}(-\mu_0 H)\right]$$
$$R_{xx,symm}^{Reverse}(\mu_0 H) = \frac{1}{2}\left[R_{xx,raw}^{Reverse}(\mu_0 H) + R_{xx,raw}^{Forward}(-\mu_0 H)\right]$$
$$R_{xy,symm}^{Forward}(\mu_0 H) = \frac{1}{2}\left[R_{xy,raw}^{Forward}(\mu_0 H) - R_{xy,raw}^{Reverse}(-\mu_0 H)\right] \tag{2}$$
$$R_{xy,symm}^{Reverse}(\mu_0 H) = \frac{1}{2}\left[R_{xy,raw}^{Reverse}(\mu_0 H) - R_{xy,raw}^{Forward}(-\mu_0 H)\right]$$

The field-dependent $R_{xx}$ signal was converted to magnetoresistance using eq. (3).

$$MR(\%) = \frac{R_{xx}(H) - R_{xx}(H=0)}{R_{xx}(H=0)} \times 100\% \qquad (3)$$

The measured transverse resistance ($R_{xy}$) was separated into the ordinary Hall effect term ($R_{OHE}$) and the anomalous Hall effect term ($R_{OHE}$) according to the following equation:

$$R_{xy} = R_{OHE} + RAHE = \frac{1}{ne}\mu_0 H + R_{AHE} \qquad (4)$$

where $n$ is the 2D carrier density, $\mu_0$ is the vacuum vacuum permeability, and $H$ is the applied magnetic field ($H\|c$). The $R_{OHE}$ term is linearly proportional to the applied magnetic field and was used to calculate the carrier density of the device. The $R_{AHE}$ term reported herein is directly proportional to the magnetization of the sample[11,12] and was obtained by subtracting the $R_{OHE}$ term from the measured $R_{xy}$ signal.

### Preparation of samples for nanoARPES

A nanothick (-10 nm) $2H$–$TaS_2$ flake was partially capped with monolayer hBN using the dry transfer method[12,55]. Next, $Fe_xC_yO_z$ was patterned onto the uncapped portion of the heterostructure, followed by annealing to 350 °C. The sample was then electrically connected with metal leads using standard nanofabrication methods at the Marvell Nanofabrication Laboratory. The fabricated Cr/Au leads were electrically connected to the ARPES Cu puck using silver epoxy, which was cured at 120 °C for 15 minutes. Lastly, the sample was heated in the ARPES chamber at 100 °C overnight immediately before the ARPES experiments.

### Nano angle-resolved photoemission spectroscopy (nanoARPES)

nanoARPES data were collected at Beamline 7.0.2 of the Advanced Light Source (ALS) on the nanoARPES endstation using Scienta Omicron R4000 hemispherical electron analyzers. The beam diameter was approximately 1 $\mu$m. All measurements were conducted at the base temperature of 19 K with linear horizontal (LH) polarization at pressures lower than $5 \times 10^{-11}$ Torr. Core-level measurements were conducted with a photon energy of 160 eV, while all other measurements were conducted with $h\nu$ = 118 eV. Data analysis was conducted using the PyARPES software package[69]. High-symmetry cuts were normalized such that the integrated intensity for each momentum and energy distribution curve are equal in the plotted range. Fermi surface cuts in the main text were normalized to ensure that the integrated intensity of the $k_x$ and $k_y$ momentum distribution curves were equivalent.

## Data availability

Source data are provided with this paper. Datasets used to generate figures in the main text and raw transport data are publicly available on Zenodo[70]. Additional data is available from the corresponding author upon request.

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

## Acknowledgements

We thank R. Murphy and M.P. Erodici for helpful discussions, and we acknowledge C. Gammer and S. Zeltmann for developing the 4D-STEM acquisition code for TitanX. We also acknowledge G.P. Hegel for help with bulk characterization. Research supported by the U.S. Department of Energy (DOE), Office of Science, Basic Energy Sciences (BES), under Award # DE-SC0025525. This research used resources of the Advanced Light Source, which is a DOE Office of Science User Facility under contract no. DE-AC02-05CH11231. Work at the Molecular Foundry, LBNL, was supported by the Office of Science, Office of Basic Energy Sciences, the U.S. Department of Energy under Contract no. DE-AC02-05CH11231. Confocal Raman spectroscopy was supported by a Defense University Research Instrumentation Program grant through the Office of Naval Research under award no. N00014-20-1-2599 (D.K.B.). Portions of this work were supported by the Gordon and Betty Moore Foundation EPiQS Initiative Award no. 10637 and the Heising Simons Faculty Fellowship. Electron microscopy was, in part, supported by the Platform for the Accelerated Realization, Analysis, and Discovery of Interface Materials (PARADIM) under NSF Cooperative Agreement no. DMR-2039380. This work made use of the Cornell Center for Materials Research (CCMR) Shared Facilities, which are supported through the NSF MRSEC Program (no. DMR- 1719875). The Thermo Fisher Spectra 300 X-CFEG was acquired with support from PARADIM, an NSF MIP (DMR-2039380) and Cornell University. K.W. and T.T. acknowledge support from JSPS KAKENHI (Grant Numbers 19H05790, 20H00354 and 21H05233). B.H.G. was supported by the University of California Presidential Postdoctoral Fellowship (PPFP) and the Schmidt Science Fellows, in partnership with the Rhodes Trust. L.S.X. was supported by an Arnold O. Beckman Postdoctoral Fellowship. S.H. acknowledges support from the Blavatnik Innovation Fellowship.

## Author contributions

S.H. and D.K.B. conceived the study. S.H., O.G. and W.Z. fabricated the samples. S.H., B.H.G., K.C.B., C.S. and J.C. imaged samples using TEM. S.H., L.X., O.G., S.H.R., S.S.F., C.J., A.B., E.R. performed nanoARPES experiments. Z.K. performed the XPS study. S.H., S.M.R. and C.O. developed the code for the virtual apertures. T.T. and K.W. provided the hBN crystals. S.H. and D.K.B. wrote the manuscript with input from all co-authors.

## Competing interests

The authors declare no competing interests.
