## [Transparent Peer Review file · Nature Communications]

Tailored topotactic chemistry unlocks heterostructures of magnetic intercalation compounds

Corresponding Author: Professor Daniel Bediako

Version 0:

Reviewer comments:

Reviewer #1

(Remarks to the Author)
Dear Editors,

the authors of the manuscript suggest the solid-state topochemistry method to tailor the TMD compounds (FexTS_2 or FexNbS_2) with a given concentration x of the intercalated ions. They demonstrate the successful synthesis of TMD heterostructures, however, there are some questions that need to be addressed before I can recommend this manuscript for publication.

1) The main chemical reaction is given by Eq. (1). The authors should clarify the methods used to remove byproducts from the reactions, i.e. Ta_2O_5 and S.

2) In Fig. 5c, units of the coercive field are not given.

3) What are Cuire and Neel temperatures of the $\text{FexTaS}_2/\text{FeyNbS}_2$ heterostructure? In general, I see no plots of temperature behavior for magnetization and susceptibility. One should expect a rather nontrivial behaviour of these characteristics in this heterostructure.

4) At the page 14, the authors write: "Given that our samples deviate from stoichiometric $\text{Fe}_{0.33}\text{NbS}_2$, we propose that these uncompensated moments arise from intercalant disorder, which is known to introduce an uncompensated spin glass phase in bulk crystals of FexNbS_2 [19,48]. The spin glass phase could become pinned by the surrounding antiferromagnetic moments, leading to the observed hysteretic behavior."

There is an alternative explanation for this effect based on metastable ferromagnetic states stabilized by magnetoelastic interactions.

[1] Magnetic order, field-induced phase transitions and magnetoresistance in the intercalated compound $\text{Fe}_{0.5}\text{TiS}_2$ NV Baranov, EM Sherokalova, NV Selezneva, AV Proshkin, AF Gubkin, Journal of Physics: Condensed Matter 25 (6), 066004 (2013).

[2] Magnetic phase transitions, metastable states, and magnetic hysteresis in the antiferromagnetic compounds $\text{Fe}_{0.5}\text{TiS}_{2-y}(\text{Se}_y)$ NV Baranov, NV Selezneva, EM Sherokalova, YA Baglaeva, Physical Review B 100 (2), 024430 (2019).

[3] Relationship between magnetoresistance behavior and magnetic states in intercalated compounds FexTiS_2 , NV Selezneva, EM Sherokalova, A Podlesnyak, M Frontzek, NV Baranov Physical Review Materials 7 (1), 014401

Of course, the TiS_2 -based materials were considered, but the mechanism of hysteresis may be common with TaS_2 -based TMD. For objectivity, please reference these works.

5) There is no plot of temperature dependence of the anomalous Hall effect. It could provide magnetic ordering temperatures for the heterostructures.

6) How effective is the proposed method for co-intercalation, when dissimilar ions are inserted into the vdW gaps?

Sincerely yours,
Referee

Reviewer #2

(Remarks to the Author)

The manuscript "Tailored topotactic chemistry unlocks heterostructures of magnetic intercalation compounds" by Husremovic et al. describes the synthesis of a class of TMD materials intercalated with Fe. Authors present a synthesis approach to form the 2D magnetic materials, propose a set of characterizations to demonstrate their structure and highlight the synthesis mechanism, and finally probe magnetic properties in fabricated devices including simple heterostructures of these materials. The study of magnetic lamellar materials is particularly attractive as it could provide materials with atomically defined interfaces to magnetic and spintronics structures. Interfaces are critical in spintronics, and lamellar materials are a particularly good fit in this regard. The work here highlights a relatively novel family of 2D magnetic materials that is attractive as it enhances an existing class of material (namely TMD, relatively well mastered) with magnetic properties. This could then extend to lateral or vertical semiconducting/ferromagnetic heterostructures in a single material platform (versus e.g. FGT, CrI₃, etc), a key requirement for envisioned future spintronics architectures.

That said, I have the following remarks related to this work, as some points are not clearly discussed:

(1) Right from the title, authors claim they "unlock" this family of materials and their heterostructures, while it is already presented in the literature (see ref 24 and 25 in particular). The main contribution here is the absence of excess oxidized Fe in the final intercalated flakes, allowing a direct interfacing between flakes. Authors should thus focus on this aspect initially and describe how their approach is able to prevent this excess of Fe in a more systematic discussion of the state of the art.

(2) Figure 1 panel c shows an incomplete intercalation in the most extreme conditions. Authors should discuss more clearly if (i) they achieved a complete intercalation (ii) if they believe it is possible at all, and (iii) in their view what would be the possible impact/limitation on magnetic and transport properties of partial intercalations.

(3) The extracted T_c of the different compounds is below 40K, making device exploitation limited to liquid He conditions. Do authors have a perspective beyond this limitation?

(4) Exploitation of 2D materials requires a perspective toward large scale, beyond exfoliated flakes. TMDs are available now thanks to CVD in particular. The application of the presented approach to such large scale TMD material appears however limited, due to diffusion mechanisms being mostly in-plane and limited to few microns (based on the presented data). Authors should discuss this scaling issue.

(5) Figure S6 present XPS data of Fe. Intriguingly, these show the coexistence of several Fe states in the resulting materials. Authors should comment more adequately on XPS, as the presence of metallic Fe or Fe oxides should be discussed clearly. Figure S11 panel (a) seems to indicate some Fe clustering on top. This point is crucial: such clustering could (i) prevent the novelty claim, see question (1) and (ii) prevent also the interpretation of transport data as magnetic particles could contribute to magnetic signals in devices.

(6) The use cases for these specific compounds are not discussed clearly.

Reviewer #3

(Remarks to the Author)

The authors of the manuscript NCOMMS-24-30362-T, titled "Tailored topotactic chemistry unlocks heterostructures of magnetic intercalation compounds," present an intriguing method for preparing intercalated transition metal dichalcogenide (TMD) devices. Unlike pristine TMDs, which are relatively easy to exfoliate, the exfoliation of their magnetic atom intercalated counterparts is challenging. Furthermore, once exfoliated, these intercalated TMDs tend to form an oxide layer on the outer surfaces. To address this issue, the authors used pristine TMDs and intercalated them in-situ by thermal annealing, employing a "reservoir" of ferromagnetic atoms ($Fe_xC_yO_z$ precursor). The intercalation is claimed to occur perpendicular to the patterned TMD layers as well as laterally through a diffusion process between the vdW coupled layers. The authors demonstrate the feasibility of their method by presenting evidence of ferromagnetic atom intercalation using transmission electron microscopy and nano-ARPES. They also discuss the possible transformation mechanism from a zerovalent metal in the precursor to a divalent metal in the vdW gap. Finally, they show their ability to create heterostructure devices and present the electrical transport characterization of a Fe/Co intercalated 2H-NbS₂/2H-TaS₂ heterostructure device.

The experimental work presented in this paper is compelling and offers a novel solution to a fundamental problem in using intercalated TMDs in devices. The concept and proof of principle justify consideration for publication in Nat. Communications. Moreover, this could potentially be a significant milestone in the utilization of magnetic TMDs in spintronic

devices, with substantial impact on the field. However, before accepting it for publication, I recommend that the claims about the diffusion of intercalates perpendicular to the TMD layers be rigorously demonstrated or omitted throughout the text. From the presented experiments, I conclude that the intercalation is exclusively lateral.

Additionally, I have a few questions and comments for the authors:

1. One drawback of this method is that intercalation will, in most cases, be non-homogeneous, leading to a gradient in intercalate concentration. The authors explicitly state in several places that the data was collected from selected areas. Therefore, this issue should be addressed in the manuscript. Have the authors attempted to quantify the possible intercalate gradients? Did they conduct a systematic test of the effect of annealing time to determine whether a gradient flattens (for a few micron-sized samples) and at what value of x this occurs? I expect x to be larger than $1/3$, raising the question of the homogeneity of $1/3$ or any other intercalation. For some intercalated TMDs, it is crucial to control x .
2. In Fig. 3c-d, it appears that the width of the ARPES spectra at the Fermi energy is comparable to or even greater than the range of values in panel d. Could the authors estimate the error bars?
3. In Fig. 4g, a device is shown with patches of $\text{Co}_x\text{C}_y\text{O}_z$ and $\text{Fe}_x\text{C}_y\text{O}_z$. How did the authors ensure there was no cross-contamination during the wet deposition of the precursor?
4. When presenting the $2\text{H-NbS}_2/2\text{H-TaS}_2$ heterostructure device (Fig. 5), no scale bar information is given. The location of the patches with the precursor for intercalation is also not indicated. How was the intercalation performed? The device appears quite long. Is there any issue with homogeneity? The TaS_2 layer appears to have some dirt on it; is the interface clean? Why do the cycling curves in Fig. 5b (the AFM regime) not coincide at 12 T? What is the physical meaning of H_c in the AFM regime?

Version 1:

Reviewer comments:

Reviewer #1

(Remarks to the Author)

In their corrected version the authors responded constructively to all my questions. I recommend to accept this well done work.

Reviewer #2

(Remarks to the Author)

Dear Editor,

I thank the Authors for their thoughtful response. I am pleased with the discussions that have emerged from my questions and the revisions made to the manuscript. Additionally, I am also positive about the way they have addressed the points raised by the other referees.

The investigation of these original magnetic lamellar materials is particularly compelling, as it offers the potential to develop devices with atomically precise interfaces for magnetic/spintronic applications. I fully support the publication of this study in your journal.

Reviewer #3

(Remarks to the Author)

The authors have provided satisfactory responses to my comments and have made the necessary adjustments to the text and figures. Therefore, I recommend the manuscript for publication.

Response to Referees

Table of Contents

Reviewer 1 comments and responses	2
Comment A1:	2
Comment A2:	2
Comment A3:	3
Comment A4:	4
Comment A5:	8
Comment A6:	9
Reviewer 2 comments and responses	11
Comment B1:	11
Comment B2:	11
Comment B3:	13
Comment B4:	14
Comment B5:	15
Comment B6:	16
Comment B7:	17
Reviewer 3 comments and responses	18
Comment C1:	18
Comment C2:	18
Comment C3:	19
Comment C4:	21
Comment C5:	21
Comment C6:	22
Comment C7i:	23
Comment C7ii:	23
Comment C7iii:	24
Comment C7iv:	24
Comment C7v:	25

Reviewer 1 comments and responses

Comment A1:

Dear Editors,

the authors of the manuscript suggest the solid-state topochemistry method to tailor the TMD compounds (FexTS₂ or FexNbS₂) with a given concentration x of the intercalated ions. They demonstrate the successful synthesis of TMD heterostructures, however, there are some questions that need to be addressed before I can recommend this manuscript for publication.

1) The main chemical reaction is given by Eq. (1). The authors should clarify the methods used to remove byproducts from the reactions, i.e. Ta₂O₅ and S.

We thank the reviewer for their comment and acknowledge the need for further clarification in our manuscript. Instead of removing the byproducts from Eq. (1), we developed two strategies to control where byproducts form, allowing us to create clean, byproduct-free regions of interest.

Byproducts form exclusively in regions where the precursors contact the TMD layers. However, intercalants can diffuse from these regions to produce clean, byproduct-free intercalated crystals. Thus, to synthesize such clean regions, we implemented two methods to control precursor placement: (1) encapsulation with hexagonal boron nitride (hBN) and (2) lithographic precursor patterning. By combining these approaches, we successfully synthesized ultra-clean Fe-intercalated TaS₂ crystals, suitable even for surface-sensitive angle-resolved photoemission spectroscopy (ARPES) measurements (Figure 3).

We **incorporated a sentence into the main text** to explicitly state that byproducts are not removed but rather can be deterministically placed:

...Thus, although byproducts of disproportionation chemistry are not eliminated, they can be confined to well-defined regions.

Comment A2:

In Fig. 5c, units of the coercive field are not given.

We thank the reviewer for bringing this to our attention. **Figure 5c was corrected accordingly** and is shown below.

Figure 5. Magnetotransport of iron-intercalated $2H\text{-NbS}_2/2H\text{-TaS}_2$ heterostructures. (a) Optical micrograph of the measured mesoscopic device. Scale bar: $5\ \mu\text{m}$. Three distinct measured regions are false-colored and labeled: R1- $\text{Fe}_{0.32(1)}\text{TaS}_2$, R2- $\text{Fe}_{0.42(4)}\text{NbS}_2 / \text{Fe}_{0.31(2)}\text{TaS}_2$ and R3- $\text{Fe}_{0.32(3)}\text{NbS}_2$. (b) Field-dependent magnetoresistance (MR) and anomalous Hall resistance (R_{AHE}) recorded in regions R1–R3. (c) Temperature-dependent coercive field (H_c) for R1–R3. For R1–R2, the average of H_c is obtained from both MR and R_{AHE} measurements. Here, H_c refers to the field where MR reaches its maximum while R_{AHE} equals zero. Conversely, for R3, H_c is determined exclusively from MR data, defining it as the field where MR exhibits its minimum (upturn). The data presented in b,c was obtained after the initial thermal and field cycling of the device.

Comment A3:

What are Cuire and Neel temperatures of the $\text{Fe}_x\text{TaS}_2/\text{Fe}_y\text{NbS}_2$ heterostructure? In general, I see no plots of temperature behavior for magnetization and susceptibility. One should expect a rather nontrivial behaviour of these characteristics in this heterostructure.

We thank the reviewer for their question. Magnetization and susceptibility were not measured because conventional magnetometry techniques are not suitable for our mesoscopic flakes with diminutive masses (approximately 10^{-12} g). Instead, electronic transport measurements were used to probe their magnetic behavior, as is typical for 2D magnetic materials. To this end, we fabricated device D1, comprising Fe_xTaS_2 (region R1), $\text{Fe}_x\text{TaS}_2/\text{Fe}_x\text{NbS}_2$ (region R2), and Fe_xNbS_2 (region R3). The ordering temperatures of R1–R3 were determined to be about 30 K, 34 K and 32 K, respectively, from the inflection in R_{xx} upon zero-field cooling (Extended Data Figure 2). These magnetic ordering temperatures were added to the main text.

Addition to the Main text:

The magnetic ordering temperatures of R1–R3 are 30 K, 34 K, and 32 K, respectively, as determined from the inflection in R_{xx} observed upon zero-field cooling (Extended Data Figure 2).

Comment A4:

At the page 14, the authors write: "Given that our samples deviate from stoichiometric $\text{Fe}_{0.33}\text{NbS}_2$, we propose that these uncompensated moments arise from intercalant disorder, which is known to introduce an uncompensated spin glass phase in bulk crystals of Fe_xNbS_2 [19,48]. The spin glass phase could become pinned by the surrounding antiferromagnetic moments, leading to the observed hysteretic behavior."

There is an alternative explanation for this effect based on metastable ferromagnetic states stabilized by magnetoelastic interactions.

[1] Magnetic order, field-induced phase transitions and magnetoresistance in the intercalated compound $\text{Fe}_{0.5}\text{TiS}_2$

NV Baranov, EM Sherokalova, NV Selezneva, AV Proshkin, AF Gubkin, *Journal of Physics: Condensed Matter* 25 (6), 066004 (2013).

[2] Magnetic phase transitions, metastable states, and magnetic hysteresis in the antiferromagnetic compounds $\text{Fe}_{0.5}\text{TiS}_2$ -y(Sey)

NV Baranov, NV Selezneva, EM Sherokalova, YA Baglaeva, *Physical Review B* 100 (2), 024430 (2019).

[3] Relationship between magnetoresistance behavior and magnetic states in intercalated compounds Fe_xTiS_2 ,

NV Selezneva, EM Sherokalova, A Podlesnyak, M Frontzek, NV Baranov *Physical Review Materials* 7 (1), 014401

Of course, the TiS_2 -based materials were considered, but the mechanism of hysteresis may be common with TaS_2 -based TMD. For objectivity, please reference these works.

We thank the reviewer for highlighting these important questions and the relevant literature. To investigate the possibility of a spin glass phase in more detail, we monitored the time-dependent magnetotransport of device D1. Our experiments are consistent with the presence of an uncompensated glassy phase exhibiting a small overall magnetization.

We also considered the possibility of a metamagnetic transition to a ferromagnetic state, as suggested by the reviewer. However, we find that an ensemble ferromagnetic state, as reported for Ti-based TMDs, is unlikely due to the absence of an anomalous Hall effect (Extended Data Figure 3c), which is directly proportional to magnetization. Nonetheless, similar to Ti-based TMDs, we observed that the virgin magnetoresistance curves for Fe_xNbS_2 are irreversible. This irreversibility suggests that the applied magnetic field may potentially stabilize the uncompensated glassy phase. Further investigation is necessary to fully understand the nature and stabilization mechanisms of this uncompensated phase, which presents an intriguing avenue for future research.

We summarized this discussion in the main text and cited the relevant Fe_xTiS_2 literature. A more expansive discussion was added to the Supplementary Information including the relevant transport figures.

Addition to the Main text:

The magnetic relaxation behavior of R3 supports the presence of glassy magnetic behavior (Supplementary Note 6, Supplementary Figure 29–31). We also note that the uncompensated phase could be stabilized through field-driven metamagnetic transitions (Supplementary Note 7, Supplementary Figure 33), which are observed in bulk crystals of Fe_xTiS_2 [1–4]. While further investigation is needed to fully elucidate the origin and nature of the uncompensated phase in R3, our data suggests that it coexists with AFM order and engenders the observed hysteretic behavior.

[1] Baranov, N. et al. Magnetic order, field-induced phase transitions and magnetoresistance in the intercalated compound $\text{Fe}_{0.5}\text{TiS}_2$. *Journal of Physics: Condensed Matter* **25**, 066004 (2013).

[2] Baranov, N. V. et al. Magnetic phase transitions, metastable states, and magnetic hysteresis in the antiferromagnetic compounds $\text{Fe}_{0.5}\text{TiS}_{2-y}\text{Se}_y$. *Phys. Rev. B* **100**, 024430 (2019).

[3] Selezneva, N. V., Baranov, N. V., Sherokalova, E. M., Volegov, A. S. & Sherstobitov, A. A. Multiple magnetic states and irreversibilities in the Fe_xTiS_2 system. *Phys. Rev. B* **104**, 064411 (6 2021).

[4] Selezneva, N. V., Sherokalova, E. M., Podlesnyak, A., Frontzek, M. & Baranov, N. V. Relationship between magnetoresistance behavior and magnetic states in intercalated compounds Fe_xTiS_2 . *Phys. Rev. Mater.* **7**, 014401 (2023).

Additions to the Supplementary Information:

Supplementary Note 6: Magnetic relaxation measured for device D1

To investigate magnetic relaxation in regions R1–R3 of device D1, we examine the changes in longitudinal resistance (R_{xx}) immediately after applying a magnetic field of 12 T (Supplementary Figure 29a). For R1 and R2, R_{xx} remains stable following magnetization (Supplementary Figure 29a). In contrast, R_{xx} of R3 evinces both gradual changes and abrupt jumps at 1.8 K (Supplementary Figure 29a,b), reflecting slow and rapid magnetic relaxation, respectively. Slow magnetic relaxation is typically linked to glassy magnetic behavior, while stochastic resistance jumps suggest the motion of pinned domain walls (DWs) driven by the applied field. Stochastic resistance jumps are also prominent in isothermal field sweeps for R3 below 10 K (Supplementary Figures 30, 31). At higher temperatures, signatures of magnetic relaxation are not evident for R3 (Supplementary Figure 29b), consistent with the expected fast relaxation times and diminishing domain pinning at elevated temperatures.

Supplementary Figure 29. Magnetic relaxation behavior of device D1. (a) Time-dependent longitudinal resistance (R_{xx}) of device regions R1–R3 at 12 T, immediately following a forward magnetic sweep from 0 T to 12 T at 1.8 K. Dashed lines in (a) serve as visual guides for the data trend. (b) Time-dependent change in the longitudinal resistance from the value recorded immediately after a forward magnetic sweep from -12 T to 12 T at 1.8 K and 30 K. For (a) and (b), $H \parallel c$ and $i \parallel ab$.

Supplementary Figure 30. Repeated measurements of field-dependent longitudinal resistance for region R3 of device D1 at 1.8 K. (a,b) Field-dependent raw (a) and symmetrized (b) longitudinal resistance (R_{xx}) of recorded for region R3 of device D1 at 1.8 K. Data from six measurements, labeled 1–6, is shown. Each measurement was obtained after ZFC protocol and started with a reverse sweep (12 T to -12 T) and ended in a forward sweep (-12 T to 12 T).

Supplementary Figure 31. Repeated measurements of field-dependent longitudinal resistance for region R3 of device D1 above 1.8 K. (a–d) Raw longitudinal resistance (R_{xx}) as a function of magnetic field for region R3 at 3 K (a), 5 K (b), 10 K (c), and 20 K (d). Each dataset, labeled 1, 3, and 6, shows measurements starting with a reverse field sweep (12 T to -12 T) and ending with a forward sweep (-12 T to 12 T). The sample was held at 12 T for 5 minutes before the reverse sweep and for 60 seconds after the forward sweep.

Supplementary Note 7: Potential field-driven metamagnetic transitions

Field-driven metamagnetic transitions from an antiferromagnetic to a metastable ferromagnetic state have been reported in bulk Fe_xTiS_2 , materials closely related to those used in device D1. These transitions can be detected by comparing initial magnetization curves after zero-field cooling with subsequent magnetic field sweeps. Large, irreversible changes in magnetoresistance indicate the presence of metamagnetic transitions. In regions R1 and R2, the resistance of the initialization curves falls within the range recorded during subsequent field sweeps, suggesting no metamagnetic changes (Supplementary Figure 33). However, in region R3, the resistance of the initialization curve lies outside this range (Supplementary Figure 33), indicating possible metamagnetic and irreversible changes when a field is applied. The absence of anomalous Hall effect (Extended Data Figure 3c), suggests that the system does not transition to an ensemble ferromagnetic state after field sweeps. This hints that the field might be stabilizing a minority magnetic phase, such as a glassy uncompensated phase.

Supplementary Figure 33. Magnetoresistance including initialization curves. (a–c) Magnetoresistance data recorded for regions R1 (a), R2 (b), and R3 (c) of device D1 at 1.8 K. The initial magnetization was obtained after zero-field cooling (ZFC). Each measurement began with a forward initialization sweep (0 to 12 T), followed by a reverse sweep (12 T to –12 T), and concluded with a forward sweep (–12 T to 12 T). Data was not symmetrized and magnetoresistance was defined as: $MR(\%) = [(R_{xx}(H) - R_{xx,init}(H = 0)) / R_{xx,init}(H = 0)] \times 100\%$, where $R_{xx,init}(H = 0)$ is the resistance of the initial magnetization curve at zero external magnetic field.

Comment A5:

There is no plot of temperature dependence of the anomalous Hall effect. It could provide magnetic ordering temperatures for the heterostructures.

We thank the reviewer for this suggestion. We added the plot of the temperature dependence of the anomalous Hall effect (AHE) at zero external field as in the Supplementary Information (Supplementary Figure 27). While the AHE signal persists above the primary magnetic ordering temperatures for regions R1 and R2, these were estimated from the inflection in resistance upon zero-field cooling (Extended Data Figure 2), and the transitions appear rather broad. The AHE data suggests that time-reversal symmetry is broken below about 40 K in R1 and R2. We incorporated this analysis into the main text and added the AHE plot in the Supplementary Information.

Addition to the Main text:

Interestingly, hysteretic magnetoresistance (MR) and remnant AHE remain observable up to 40 K for regions R1–R2 (Figure 5c, Extended Data Figure 3a,b, Supplementary Figure 27) consistent with the onset of broken time reversal symmetry at temperatures higher than those estimated from the inflection in resistance upon zero-field cooling.

Addition to the Supplementary Information:

Supplementary Figure 27. Remnant anomalous Hall resistance for device D1. Temperature-dependence of the remnant anomalous Hall resistance R_{AHE} ($H = 0$) for regions R1 and R2 of device D1.

Comment A6:

How effective is the proposed method for co-intercalation, when dissimilar ions are inserted into the vdW gaps?

Thank you for the insightful question. In the main text, we demonstrated that patterning dissimilar precursors on stacked TaS₂ flakes results in a co-intercalated heterostructure, where each flake contains a higher concentration of the metal patterned directly onto it (Figure 4f–k).

Additionally, we found that dissimilar ions can be co-intercalated into a single TMD flake by patterning both Co and Fe precursors on separate ends of a TaS₂ flake, followed by vacuum annealing. Cross-sectional STEM-EDS analysis of this sample revealed uniform co-intercalation. This data has been included in the Supplementary Information and referenced in the main text. However, while these experiments demonstrate the viability of co-intercalation, a detailed investigation of co-intercalated structures is beyond the scope of this work and represents a promising direction for future research.

Addition to the Main text:

...This observed intercalant segregation distinguishes these heterostructures from bulk crystals grown with solid-state methods and singular flakes intercalated with two precursors (Supplementary Figure 23), where multiple metal intercalants disperse uniformly throughout the host lattice.

Addition to the Supplementary Information:

Supplementary Figure 23. Co-intercalation of Co and Fe in a singular $2H\text{-TaS}_2$ flake. (a) Scanning electron microscopy (SEM) image of a $2H\text{-TaS}_2$ flake (outlined with dashed white lines) with a $\text{Co}_x\text{C}_y\text{O}_z$ precursor patterned on the top (false-colored in violet) and a $\text{Fe}_x\text{C}_y\text{O}_z$ precursor patterned on the bottom (false-colored in red). Scale bar: $5\ \mu\text{m}$. (b) Low-frequency Raman spectrum of the flake region between the two metal precursors. (c) Cumulative STEM-EDS spectrum of the cross-sectioned flake region between the two metal precursors. Cross-sectioning was performed using a focused ion beam, and the STEM-EDS data was collected $\perp c$ axis of $2H\text{-TaS}_2$.

Reviewer 2 comments and responses

Comment B1:

The manuscript “Tailored topotactic chemistry unlocks heterostructures of magnetic intercalation compounds” by Husremovic et al. describes the synthesis of a class of TMD materials intercalated with Fe. Authors present a synthesis approach to form the 2D magnetic materials, propose a set of characterizations to demonstrate their structure and highlight the synthesis mechanism, and finally probe magnetic properties in fabricated devices, including simple heterostructures of these materials. The study of magnetic lamellar materials is particularly attractive as it could provide materials with atomically defined interfaces to magnetic and spintronics structures. Interfaces are critical in spintronics, and lamellar materials are a particularly good fit in this regard. The work here highlights a relatively novel family of 2D magnetic materials that is attractive as it enhances an existing class of material (namely TMD, relatively well mastered) with magnetic properties. This could then extend to lateral or vertical semiconducting/ferromagnetic heterostructures in a single material platform (versus e.g. FGT, CrI₃, etc), a key requirement for envisioned future spintronics architectures.

That said, I have the following remarks related to this work, as some points are not clearly discussed:

We sincerely appreciate the reviewer’s encouraging feedback on our work and their valuable suggestions for improving the manuscript.

Comment B2:

Right from the title, authors claim they “unlock” this family of materials and their heterostructures, while it is already presented in the literature (see ref 24 and 25 in particular). The main contribution here is the absence of excess oxidized Fe in the final intercalated flakes, allowing a direct interfacing between flakes. Authors should thus focus on this aspect initially and describe how their approach is able to prevent this excess of Fe in a more systematic discussion of the state of the art.

We thank the reviewer for highlighting the need to clarify the novelty of our study. In previous works (refs 24 and 25), heterostructures were made from thick flakes (approximately 100 nm) cleaved from bulk crystals of intercalated TMDs. A major limitation of these experiments is that the cleaved flakes are typically thick (~100 nm) due to the strong interlayer interactions imparted by the intercalants. In contrast, we cleave non-intercalated TMDs, which can be obtained at the atomically thin limit. This allows us to control the dimensionality of the samples, which is critical for miniaturization and optimization in device applications.

Moreover, *interfaces* between flakes cleaved from bulk crystals tend to show disorder. This disorder does not arise from oxidized Fe, as the bulk crystals are grown using zero-valent Fe precursors. Instead, the disorder comes from native oxides (e.g., Ta₂O₅) of the host lattice (TaS₂). Additionally, cleaving bulk intercalated crystals often results in surfaces with dissimilar intercalant coverage [see *Chemistry of Materials* **35**, 7239–7251 (2023)]. This, in turn, causes termination disorder and surface inhomogeneities that hinder the formation of atomically sharp interfaces. In

contrast, our synthetic approach eliminates native oxides or termination disorder at the interface of stacked flakes, enabling the formation of sharper and electronically more defined heterointerfaces.

So, while we do use metal oxides as intercalation precursors, these do not affect the heterointerfaces between the stacked flakes. Additionally, we developed two strategies to keep the top interfaces clean and free from contamination in regions of interest. First, we can cap regions of interest with hexagonal boron nitride (hBN) to prevent the precursor from contacting the TMD in hBN-capped areas. Second, we use nanolithographic patterning to selectively place the precursors in designated regions. Since intercalants diffuse from areas in direct contact with the precursor, intercalated TMDs can also form away from these regions, resulting in products with pristine top interfaces. By combining hBN capping and regioselective precursor patterning, we successfully created ultraclean top interfaces suitable for highly surface-sensitive techniques like angle-resolved photoemission spectroscopy (ARPES).

To summarize, our work enables the creation of heterostructures with (1) controllable dimensionality, (2) pristine heterointerfaces, and (3) ultraclean surfaces via deterministic precursor placement. We emphasized the novelty of our intercalated structures and more clearly positioned our work within the context of previous literature in the introduction of the paper.

Main text, Introduction:

...Even established techniques for fabricating heterostructures of layered crystals, such as mechanical exfoliation followed by vdW assembly, are not viable for intercalated crystals: the strong interlayer interactions imparted by the intercalants pose significant challenges in isolating thin crystals [1,2] which are integral components of heterostructures. Thus, achieving dimensionality control in heterostructures requires moving beyond the use of flakes exfoliated from bulk intercalated materials. Such exfoliated flakes also inherently exhibit inhomogeneous intercalant distribution on cleaved surfaces [3] and form surface native oxides, hindering the creation of atomically sharp heterointerfaces. Consequently, prior work on vdW assembly of intercalated TMDs has been limited to fabricating magnetic tunnel junctions with incidental oxide formation at poorly defined and controlled heterointerfaces [1, 2].

Here, we establish a synthetic framework to create pristine heterostructures comprising magnetic TMD intercalation compounds of 2H-TaS₂ and 2H-NbS₂ containing Fe and Co. These low-dimensional heterostructures exhibit tunable dimensionality, atomically sharp heterointerfaces with modular symmetry and structure, and ultraclean surfaces via deterministic precursor placement...

[1] Yamasaki, Y. et al. Exfoliation and van der Waals heterostructure assembly of intercalated ferromagnet Cr_{1/3}TaS₂. *2D Materials* **4**, 041007 (2017).

[2] Arai, M. et al. Construction of van der Waals magnetic tunnel junction using ferromagnetic layered dichalcogenide. *Applied Physics Letters* **107**, 103107 (2015).

[3] Xie, L. S. et al. Comparative Electronic Structures of the Chiral Helimagnets Cr_{1/3}NbS₂ and Cr_{1/3}TaS₂. *Chemistry of Materials* **35**, 7239–7251 (2023).

Comment B3:

Figure 1 panel c shows an incomplete intercalation in the most extreme conditions. Authors should discuss more clearly if (i) they achieved a complete intercalation (ii) if they believe it is possible at all, and (iii) in their view what would be the possible impact/limitation on magnetic and transport properties of partial intercalations.

We thank the reviewer for the insightful questions. We address their comments one-by-one below:

- (i) Complete intercalation was *not* achieved in sample S3. This is evident from its Raman spectrum, which evinces broad low-frequency Fe modes, characteristic of disordered intercalants, and a low intercalation amount ($\text{Fe}/\text{Ta} < 0.2$). We have now included the Raman data for samples S1–S3 in the Supplementary Information.

Addition to the Supplementary Information:
Supplementary Figure 3. Raman of samples S1–S3. Raman of samples S1–S3 and pristine, reference $2H\text{-TaS}_2$ flake that was not chemically treated. Samples S1–S3 were treated with $\text{Fe}(\text{CO})_5$ in acetone and subsequently vacuum annealed at $100\text{ }^\circ\text{C}$, $200\text{ }^\circ\text{C}$ and $350\text{ }^\circ\text{C}$, respectively. Thickness of S1–S3 is 12 nm, 7.8 nm, and 23 nm, respectively. S3 is the only sample whose Raman spectra exhibit Fe-related modes and peak shifts in comparison to the pristine sample.

- (ii) Sample S3, being relatively thick (23 nm), did not form an ordered intercalation compound after 30 minutes of heating at $350\text{ }^\circ\text{C}$. However, under the same annealing conditions, thinner samples typically show a higher intercalant concentration and degree of intercalant ordering. This was demonstrated in our previous study [1], where we investigated a 12 nm sample treated with $\text{Fe}(\text{CO})_5$ /toluene and subsequently annealed at $350\text{ }^\circ\text{C}$ for 30 minutes (please see Review Figure 1 below). The Raman spectrum of this sample shows sharp low-frequency modes (labeled Fe^*), indicating an ordered $\sqrt{3} \times \sqrt{3}$ lattice. This ordered intercalation was further confirmed by atomic-resolution differential phase contrast micrographs (Review Figure 1b).

Review Figure 1. Raman spectrum of a 12-nm Fe_xTaS_2 sample. (b) Cross-sectional differential phase contrast (DPC) image of a cross-sectioned sample from (a). Data is taken from Ref [1].

[1] Husremović S. et al. Hard ferromagnetism down to the thinnest limit of iron-intercalated tantalum disulfide. *Journal of the American Chemical Society* **144**, 12167–12176 (2022).

- (iii) Indeed, partial intercalation in off-stoichiometric or disordered crystals can create inhomogeneities in magnetic interactions, leading to emergent behaviors such as glassy magnetism. However, in disordered samples, it is essential to carefully correlate the structure with the observed magnetoelectric properties to derive accurate structure-property relationships. To this end, we routinely characterize the measurement channel of our mesoscopic devices with Raman spectroscopy and a suite of transmission electron microscopy techniques.

Comment B4:

The extracted T_c of the different compounds is below 40K, making device exploitation limited to liquid He conditions. Do authors have a perspective beyond this limitation?

We thank the reviewer for raising this question. One approach to raising the ordering temperatures is to increase the spin-orbit coupling (SOC) in our materials. This can be achieved by substituting sulfides with selenides, co-intercalating lanthanides, or interfacing our heterostructures with heavy compounds such as Bi-based 2D materials. These strategies could help overcome the limitation of low T_c and broaden the operational temperature range of our devices. For example, we have seen magnetic phenomena up to room temperature in this family of materials when we substitute Se for S [*J. Am. Chem. Soc.* 2023, 145, 20041]. However, contrary to popular opinion, higher temperatures are not necessarily needed for all applications. There is a real need for *cryogenic* memories to interface with quantum computing architectures at low temperatures. These will avoid issues related to increased thermal load, noise, and signal integrity that would otherwise be problematic for interfacing a qubit at cryogenic temperatures via cables to room-temperature memories and other electronic elements.

Comment B5:

Exploitation of 2D materials requires a perspective toward large scale, beyond exfoliated flakes. TMDs are available now thanks to CVD in particular. The application of the presented approach to such large scale TMD material appears however limited, due to diffusion mechanisms being mostly in-plane and limited to few microns (based on the presented data). Authors should discuss this scaling issue.

We appreciate the reviewer's comment and recognize the importance of scaling 2D materials beyond exfoliated flakes. While the synthetic approach presented here yields micron-sized samples, the insights gained can help identify candidates for more complex scale-up. Additionally, the diffusion mechanisms established in our study can inform future scale-up efforts.

Additional experiments revealed that vertical diffusion, in addition to lateral diffusion, occurs readily within a single TMD flake but is slower across moiré interfaces. This knowledge could be applied to design a CVD-based synthesis. For example, the workflow could involve synthesizing a wafer-scale TMD, intercalating it in a precursor-rich atmosphere, and repeating this process for each flake in the intercalated heterostructure. In this process, efficient vertical diffusion would enable fast and efficient wafer-scale intercalation.

We included a brief discussion of vertical diffusion in the main text, along with evidence of Fe vertical diffusion in a $2H$ -TaS₂ flake in the Supplementary Information.

Addition to the Supplementary Information:
Supplementary Figure 14. Vertical diffusion of intercalants. (a) Optical micrograph of a $2H$ -TaS₂ flake with a $Fe_xC_yO_z$ precursor patterned in the middle. The precursor was drop-casted from a 0.74 M solution of $Fe(CO)_5$ in isopropanol. The precursor is outlined in a red dashed line and false-colored red. The flake is outlined in a dashed white line. Scale bar: 10 μm . (b) Raman spectra of the sample from (a) in the region marked with a blue circle after annealing for 30 minutes at 250 $^{\circ}\text{C}$. The Fe-related mode is evident at 141 cm^{-1} , and the A_{1g} mode is observed at 394 cm^{-1} ; both spectral features are characteristic of intercalation.

Addition to the Main text:

...We first investigate the vertical diffusion of intercalants by patterning precursors at the center of 2H-TaS₂ flakes, which prevents lateral diffusion from the edges and confines the movement of ions to the out-of-plane direction. The successful intercalation observed in these samples confirms that vertical diffusion takes place during the intercalation process (Supplementary Figure 14).

Comment B6:

Figure S6 present XPS data of Fe. Intriguingly, these show the coexistence of several Fe states in the resulting materials. Authors should comment more adequately on XPS, as the presence of metallic Fe or Fe oxides should be discussed clearly.

The Fe precursors studied contain varying proportions of Fe species, as revealed by XPS and EELS analyses. Fe_xC_yO_z precursor films includes both Fe²⁺ and Fe³⁺ in varying ratios depending on the solvent, while evaporated Fe metal contains a small amount of Fe⁰ alongside oxidized Fe species. Despite these differences, atomic resolution EELS and confocal Raman data support that Fe intercalated in TaS₂ consistently remains in the Fe²⁺ state, regardless of the precursor (Figure 1d,e and Supplementary Figure 1). A detailed discussion of Fe oxidation states in the precursors was added to the Supplementary Information.

Addition to the Supplementary Information:

Supplementary Note 2: Oxidation state of Fe precursors

Fe oxidation states in various precursors were analyzed using energy-loss electron spectroscopy (EELS) and X-ray photoelectron spectroscopy (XPS), revealing distinct proportions of Fe⁰, Fe²⁺, and Fe³⁺. Oxidized Fe species predominated across all samples, with only evaporated Fe films showing a minor proportion of Fe⁰, amounting to less than 10 % of the total.

Fe_xC_yO_z precursors contain a mixture of Fe³⁺ and Fe²⁺, with the ratio influenced by the solvent used during film preparation. Solvents with higher oxygen and water content, such as acetone and isopropanol, produced Fe_xC_yO_z with the highest proportion of Fe³⁺. Specifically, we observed nearly 100 % Fe³⁺ using acetone (Figure 1e, Supplementary Figure 6) and 80 % Fe³⁺ using isopropanol (Supplementary Figure 7a,b). In contrast, films prepared in dry, degassed toluene contained about 60 % Fe³⁺ (Supplementary Figure 7c,d).

Comment B7:

Figure S11 panel (a) seems to indicate some Fe clustering on top. This point is crucial: such clustering could (i) prevent the novelty claim, see question (1) and (ii) prevent also the interpretation of transport data as magnetic particles could contribute to magnetic signals in devices.

We thank the reviewer for the insightful question. As noted, Fe nanoparticles can cluster on the surface of TaS₂ when a Fe_xC_yO_z precursor is deposited (Figure S11). However, we would like to clarify several key points that address both the novelty of our approach and the reliability of our transport data.

First, the presence of precursors on the top surface does not affect the atomically sharp heterointerfaces between stacked flakes. This ability to create pristine, atomically sharp interfaces is central to the novelty of our approach, allowing us to engineer interfacial coupling between dissimilar magnetic materials e.g., Fe_xTaS₂/Co_yTaS₂ or Fe_xTaS₂/Fe_xNbS₂.

Second, it is important to note that as far as we have been able to measure, the Fe_xC_yO_z precursors are insulating and do not contribute to the electrical signal in transport measurements (the intercalated TMDs are highly metallic and dominate the transport response). Nevertheless, we have implemented two strategies to eliminate any potential influence from the precursors and ensure ultraclean surfaces.

1. We partially encapsulate the TMDs with hexagonal boron nitride (hBN), which prevents direct contact between the precursor and the TMD surface (Figure S11). Despite this, intercalation still occurs via ion diffusion from the exposed regions of the flake. This approach allows us to probe only the hBN-covered regions during transport, excluding any influence from the precursor film, clustering etc.
2. We developed a precise patterning method to selectively place the precursor, keeping key regions clean. By combining hBN capping with this patterning technique, we produce samples where the transport channels are pristine and even with ultraclean surfaces, making them ideally suited for extremely surface-sensitive measurements like ARPES (Figure 3).

Reviewer 3 comments and responses

Comment C1:

The authors of the manuscript NCOMMS-24-30362-T, titled “Tailored topotactic chemistry unlocks heterostructures of magnetic intercalation compounds,” present an intriguing method for preparing intercalated transition metal dichalcogenide (TMD) devices. Unlike pristine TMDs, which are relatively easy to exfoliate, the exfoliation of their magnetic atom intercalated counterparts is challenging. Furthermore, once exfoliated, these intercalated TMDs tend to form an oxide layer on the outer surfaces. To address this issue, the authors used pristine TMDs and intercalated them in-situ by thermal annealing, employing a “reservoir” of ferromagnetic atoms ($\text{Fe}_x\text{C}_y\text{O}_z$ precursor). The intercalation is claimed to occur perpendicular to the patterned TMD layers as well as laterally through a diffusion process between the vdW coupled layers. The authors demonstrate the feasibility of their method by presenting evidence of ferromagnetic atom intercalation using transmission electron microscopy and nano-ARPES. They also discuss the possible transformation mechanism from a zerovalent metal in the precursor to a divalent metal in the vdW gap. Finally, they show their ability to create heterostructure devices and present the electrical transport characterization of a Fe/Co intercalated 2H-NbS₂/2H-TaS₂ heterostructure device. The experimental work presented in this paper is compelling and offers a novel solution to a fundamental problem in using intercalated TMDs in devices. The concept and proof of principle justify consideration for publication in Nat. Communications. Moreover, this could potentially be a significant milestone in the utilization of magnetic TMDs in spintronic devices, with substantial impact on the field.

We thank the reviewer for their positive feedback and recognition of the significance and potential impact of our work.

Comment C2:

However, before accepting it for publication, I recommend that the claims about the diffusion of intercalates perpendicular to the TMD layers be rigorously demonstrated or omitted throughout the text. From the presented experiments, I conclude that the intercalation is exclusively lateral.

We thank the reviewer for highlighting the need for further experimental validation of vertical diffusion. In response, we conducted additional experiments to investigate this process. By patterning a $\text{Fe}_x\text{C}_y\text{O}_z$ precursor at the center of a 2H-TaS₂ flake, we effectively prevented lateral diffusion from the edges, restricting intercalation to the vertical direction. Raman spectroscopy confirmed successful intercalation beneath the precursor, as indicated by peak shifts and the emergence of Fe-related low-frequency modes. Moreover, the presence of a singular A_{1g} peak at 394 cm^{-1} , characteristic of intercalated TaS₂, confirms that all layers beneath the precursor were intercalated. This result can only be explained by vertical diffusion from the top interface through the layers. Such vertical diffusion has been observed in MoS₂ with Li⁺ intercalants [1], whose ionic radius is comparable to that of Fe²⁺ [2]. We revised the main text to include this discussion and included the corresponding Raman data in the Supplementary Information.

[1] Zhang, J. *et al.* Reversible and selective ion intercalation through the top surface of few-layer MoS₂. *Nat Commun* **9**, 5289 (2018)

[2] Shannon, R. D. Revised effective ionic radii and systematic studies of interatomic distances in halides and chalcogenides. *Acta Crystallographica Section A* **32**, 751–767 (1976).

Addition to the Main text:

...We first investigate the vertical diffusion of intercalants by patterning precursors at the center of 2H-TaS₂ flakes, which prevents lateral diffusion from the edges and confines the movement of ions to the out-of-plane direction. The successful intercalation observed in these samples confirms that vertical diffusion takes place during the intercalation process (Supplementary Figure 14).

Addition to the Supplementary Information:

Supplementary Figure 14. Vertical diffusion of intercalants. (a) Optical micrograph of a 2H-TaS₂ flake with a Fe_xC_yO_z precursor patterned in the middle. The precursor was drop-casted from a 0.74 M solution of Fe(CO)₅ in isopropanol. The precursor is outlined in a red dashed line and false-colored red. The flake is outlined in a dashed white line. Scale bar: 10 μm. (b) Raman spectra of the sample from (a) in the region marked with a blue circle after annealing for 30 minutes at 250 °C. The Fe-related mode is evident at 141 cm⁻¹, and the A_{1g} mode is observed at 394 cm⁻¹; both spectral features are characteristic of intercalation.

Comment C3:

One drawback of this method is that intercalation will, in most cases, be non-homogeneous, leading to a gradient in intercalate concentration. The authors explicitly state in several places that the data was collected **from selected areas**. Therefore, this issue should be addressed in the manuscript. Have the authors attempted to quantify the possible intercalate gradients?

Thank you for the insightful question. We agree that characterizing intercalation inhomogeneities and gradients is crucial for accurately establishing structure-property relationships. To address this,

we made significant efforts to quantify these gradients using a combination of transmission electron microscopy (TEM), Raman spectroscopy, and angle-resolved photoemission spectroscopy (ARPES). A summary of our approach is provided below.

In **Figure 2**, we examined the composition and structural gradients in samples prepared with patterned precursors. We used scanning transmission electron microscopy (STEM) combined with energy dispersive X-ray spectroscopy (EDS) to evaluate gradients in intercalant concentration. This analysis was complemented with selected area electron diffraction (SAED), which uncovered gradients in intercalant ordering and superstructures.

In **Figure 3**, we correlated the size of hole pockets measured by ARPES with intercalation gradients identified through Raman spectroscopy (Supplementary Figure 20).

For **device D1**, we collected Raman spectra from various regions and performed electrical measurements in areas with minimal spectral variations, indicating intercalation homogeneity. We also cross-sectioned D1 in the measured regions to obtain nanoscale compositional data using STEM-EDS. This analysis was used to assign the device composition referred to in the main text and marked in the Extended Data Figure 3.

In the main text, we added a brief discussion to emphasize the importance of characterizing intercalation homogeneity to establish accurate structure-property relationships. Additionally, Raman spectra from different areas within regions R1 to R3 of device D1 were included in the Supplementary Information to demonstrate the homogeneity of the measured regions.

Addition to the Main text:

...Such intercalant gradients can significantly affect magneto-electronic behavior, making their characterization crucial for establishing accurate structure-property correlations.

Addition to the Supplementary Information:

Supplementary Figure 24. Homogeneity of device D1. Raman spectra from different areas within regions R1–R3 of device D1.

Comment C4:

Did they conduct a systematic test of the effect of annealing time to determine whether a gradient flattens (for a few micron-sized samples) and at what value of x this occurs? I expect x to be larger than $1/3$, raising the question of the homogeneity of $1/3$ or any other intercalation. For some intercalated TMDs, it is crucial to control x .

We appreciate the reviewer's constructive question. Indeed, we have conducted such annealing studies on Fe intercalation in low-dimensional $2H$ -NbSe₂. However, we think those studies lie somewhat outside the scope of this particular work and will be published separately. We hope our responses to the referee's previous comments on homogeneity will suffice for this work. As follows, here we will provide a summary of our overall findings from the other ongoing work on annealing studies.

Under our standard annealing conditions (350°C), we observed that the intercalation gradient flattens with prolonged annealing times. Thinner flakes (< 10 nm) and smaller samples (~ 5 x 10 μm) typically show reduced intercalation gradients and are able to achieve homogeneity. These gradients often exhibit a dominant $\sqrt{3} \times \sqrt{3}$ superlattice, occasionally accompanied by a weaker 2×2 mode, as demonstrated in the Raman spectra of our ARPES sample (Supplementary Figure 20).

Furthermore, the samples tend to appear homogeneous across a few-micron region, allowing the use of high-resolution probes—such as confocal Raman, (S)TEM, nanoARPES, and transport measurements—to thoroughly investigate the structure and properties of each area. Additionally, by carefully designing the precursor placement and flake geometry, we can achieve uniformity in typical device sizes (~10 × 5 μm).

Comment C5:

In Fig. 3c-d, it appears that the width of the ARPES spectra at the Fermi energy is comparable to or even greater than the range of values in panel d. Could the authors estimate the error bars?

Thank you for pointing this out. We included error bars in Figure 3d, which represent the standard errors for the center positions of Lorentzian peaks corresponding to the main hole pocket around Γ . The revised figure is shown below.

Figure 3. Spatially mapping the band structure of ultra-clean intercalated heterostructures with nanoARPES. (a) A schematic depicting the nanoARPES experiment conducted on a Fe_xTaS₂ encapsulated with monolayer hBN. The sample was prepared by selectively applying Fe_xC_yO_z onto the area of 2H-TaS₂ not covered by hBN, followed by vacuum annealing. (b,c) Normalized ARPES Fermi surface (b) and ARPES band dispersion along the Γ -K direction (c) of a hBN/Fe_xTaS₂ heterostructure at a 4 μm distance from the patterned Fe_xC_yO_z precursor. In (c), the Fermi wavevector (k_F) is marked, denoting where the band forming the hole pocket around Γ intersects the Fermi level (E_F). This band is marked by a white dashed line. Data in (b-c) was obtained at 19 K with $h\nu = 118$ eV and linear horizontal (LH) polarization. (d) k_F along the Γ -K direction extracted from ARPES band-dispersions obtained at different distances from the Fe_xC_yO_z precursor. The k_F values were obtained by fitting the momentum distribution curves (MDCs) at the Fermi level (E_F) to Lorentzians. Error bars represent the standard errors for the center positions of Lorentzian peaks corresponding to the main hole pocket around Γ . The dashed gray line is included as a visual guide. (e) Schematic of the band forming the hole pocket at Γ as the distance from the precursor increases from spot 1 to 4. (f) Qualitative d-orbital splitting diagram for the trigonal prismatic Ta center in Fe_xTaS₂. A dashed electron in the d_{z^2} orbital denotes additional electron filling upon Fe intercalation, concomitant with charge transfer to 2H-TaS₂. (g) Qualitative representation of the density of states of Fe-intercalated 2H-TaS₂.

Comment C6:

In Fig. 4g, a device is shown with patches of Co_xC_yO_z and Fe_xC_yO_z. How did the authors ensure there was no cross-contamination during the wet deposition of the precursor?

To ensure no cross-contamination of precursors, we employ a *two-step patterning process*.

First, we deposit the Fe_xC_yO_z precursor. The entire chip is initially covered with PMMA e-beam resist polymer, and e-beam lithography is used to define the region for Fe_xC_yO_z deposition. After developing the resist in this area, we deposit a solution of Fe(CO)₅/isopropanol, followed by an acetone lift-off process, leaving the Fe_xC_yO_z patch. During these steps, the region designated for Co_xC_yO_z is protected by PMMA, preventing any Fe_xC_yO_z contamination.

Second, we repeat this procedure for the $\text{Co}_x\text{C}_y\text{O}_z$ deposition using $\text{Co}_2(\text{CO})_8$ /isopropanol. During this step, the $\text{Fe}_x\text{C}_y\text{O}_z$ patch is shielded by a PMMA overlay, ensuring no cross-contamination between the two precursors.

Comment C7i:

When presenting the 2H-NbS₂/2H-TaS₂ heterostructure device (Fig. 5), no scale bar information is given.

We thank the reviewer for pointing this out. A scale bar has been added to the device micrograph (see below).

Figure 5. Magnetotransport of iron-intercalated 2H-NbS₂/2H-TaS₂ heterostructures. (a) Optical micrograph of the measured mesoscopic device. Scale bar: 5 μm. Three distinct measured regions are false-colored and labeled: R1-Fe_{0.32(1)}TaS₂, R2-Fe_{0.42(4)}NbS₂ / Fe_{0.31(2)}TaS₂ and R3-Fe_{0.32(3)}NbS₂. (b) Field-dependent magnetoresistance (MR) and anomalous Hall resistance (R_{AHE}) recorded in regions R1–R3. (c) Temperature-dependent coercive field (H_c) for R1–R3. For R1–R2, the average of H_c is obtained from both MR and R_{AHE} measurements. Here, H_c refers to the field where MR reaches its maximum while R_{AHE} equals zero. Conversely, for R3, H_c is determined exclusively from MR data, defining it as the field where MR exhibits its minimum (upturn). The data presented in b,c was obtained after the initial thermal and field cycling of the device.

Comment C7ii:

The location of the patches with the precursor for intercalation is also not indicated. How was the intercalation performed?

The intercalation for D1 was performed by depositing a $\text{Fe}_x\text{C}_y\text{O}_z$ film deposited from a $\text{Fe}(\text{CO})_5$ / toluene solution, followed by annealing the heterostructure at 350 °C for 1.5 hours. In this case, patterning of the precursor film was not used. The heterostructure used to fabricate D1 was first introduced in Figure 4, with synthesis details provided in the main text:

“To synthesize clean Fe_xTaS_2 / Fe_yNbS_2 interfaces, we prepare $2H-TaS_2 / 2H-NbS_2$ heterostructures using vdW assembly, deposit a $Fe_xC_yO_z$ film from $Fe(CO)_5$ / toluene solution, and anneal the heterostructure at $350\text{ }^\circ\text{C}$ for 1.5 hours (Figure 4a).”

Comment C7iii:

The device appears quite long. Is there any issue with homogeneity?

We investigated the homogeneity of the device using confocal Raman spectroscopy (which is sensitive to the intercalant superlattice structure) and cross-sectional STEM-EDS. As noted in the main text, the compositions of the three measured regions based on STEM-EDS are R1: $Fe_{0.32(1)}TaS_2$, R2: $Fe_{0.42(4)}NbS_2/Fe_{0.31(2)}TaS_2$, and R3: $Fe_{0.32(3)}NbS_2$. Thus, Fe content in TaS_2 appears consistent across regions, whereas Fe concentration in NbS_2 shows greater regional variability.

We also performed Raman spectroscopy across different areas of R1–R3 to assess the homogeneity within each region. The Raman spectra were consistent within each region, and these spectra have now been included in the Supplementary Information.

Addition to the Supplementary Information:

Supplementary Figure 24. Homogeneity of device D1. Raman spectra from different areas within regions R1–R3 of device D1.

Comment C7iv:

The TaS_2 layer appears to have some dirt on it; is the interface clean?

We designed the device contact placement to ensure that the measurement region excludes any signal from the area of the TaS_2 area with this residue. Moreover, we confirmed that the heterointerface between Fe_xTaS_2 and Fe_xNbS_2 is atomically sharp, as shown in Figure 4d,e. Lastly,

the transport trends in R1–R3 align with the expected behavior of metallic intercalated TMDs, confirming that we measured the intrinsic properties of our samples.

Comment C7v:

Why do the cycling curves in Fig. 5b (the AFM regime) not coincide at 12 T?

Thank you for the insightful question. We examined the magnetic behavior of our samples and found that the AFM Fe_xNbS_2 regions exhibit time-dependent magnetic relaxation at low temperatures, causing the measured resistance to vary when held at a fixed temperature and magnetic field (new data shown below). These insights explain the offset of cycling curves measured in the AFM regions at 12 T. During data collection for Fig. 5b, the wait times at 12 T differed between the reverse (12 T to -12 T) and forward (-12 T to 12 T) sweeps. As a result, due to the magnetic relaxation of these regions, the curves at 12 T did not overlap in the raw data. Consequently, in the symmetrized data presented in Fig. 5b, the cycling curves also do not coincide at 12 T. These slow relaxations of the AFM region are suggestive of glassy magnetization behavior, which has also been observed for bulk $\text{Fe}_{1/3}\text{NbS}_2$ [Nature Phys, 17, 525 (2021)] and may enable ultralow-power electrical switching of magnetization in these materials [Sci. Adv. 7 eabd8452 (2021)].

A summary of the relaxation behavior has now been included in the main text, while a comprehensive discussion of magnetic relaxation and its effects on both the raw and symmetrized data is now provided in the Supplementary Information.

Addition to the Main text:

The magnetic relaxation behavior of R3 supports the presence of glassy magnetic behavior (Supplementary Note 6, Supplementary Figure 29–31).

Addition to the Supplementary Information:

Supplementary Note 6: Magnetic relaxation measured for device D1

To investigate magnetic relaxation in regions R1–R3 of device D1, we examine the changes in longitudinal resistance (R_{xx}) immediately after applying a magnetic field of 12 T (Supplementary Figure 29a). For R1 and R2, R_{xx} remains stable following magnetization (Supplementary Figure 29a). In contrast, R_{xx} of R3 evinces both gradual changes and abrupt jumps at 1.8 K (Supplementary Figure 29a,b), reflecting slow and rapid magnetic relaxation, respectively. Slow magnetic relaxation is typically linked to glassy magnetic behavior, while stochastic resistance jumps suggest the motion of pinned domain walls (DWs) driven by the applied field. Stochastic resistance jumps are also prominent in isothermal field sweeps for R3 below 10 K (Supplementary Figures 30, 31). At higher temperatures, signatures of magnetic relaxation are not evident for R3 (Supplementary

Figure 29b), consistent with the expected fast relaxation times and diminishing domain pinning at elevated temperatures.

Supplementary Figure 29. Magnetic relaxation behavior of device D1. (a) Time-dependent longitudinal resistance (R_{xx}) of device regions R1–R3 at 12 T, immediately following a forward magnetic sweep from 0 T to 12 T at 1.8 K. Dashed lines in (a) serve as visual guides for the data trend. (b) Time-dependent change in the longitudinal resistance from the value recorded immediately after a forward magnetic sweep from -12 T to 12 T at 1.8 K and 30 K. For (a) and (b), $H \parallel c$ and $i \parallel ab$.

Examining magnetic relaxation is essential for understanding the R_{xx} hysteresis shape observed in isothermal field sweeps for R3. Specifically, this relaxation may cause the resistance states at 12 T to differ between forward and reverse sweeps, as shown in measurements 1, 5, and 6 in the Supplementary Figure 30a. In these cases, the sample was held at 12 T for over 30 minutes before the reverse sweep from 12 T to -12 T (data before the sweep was not recorded) and only 60 seconds after the forward sweep (-12 T to 12 T). This difference in hold times, and consequently in magnetic relaxation, leads to the variation in resistance states at 12 T for different sweep directions. In contrast, for measurements 2–4, the sample was held at 12 T for 60 seconds before the reverse sweep and after the forward sweep, resulting in nearly identical resistance at 12 T for both directions (Supplementary Figure 30a). It is important to note that if resistance states differ at 12 T between sweeps, the hysteresis appears open (discontinuous) after the symmetrization procedure (Supplementary Figure 30b), which mathematically equates the sweep directions. Moreover, varying relaxation times at 12 T between sweeps cause vertical offsets, altering the crossing point between sweeps and leading to asymmetric hystereses (Supplementary Figure 30a). Such asymmetry is not seen in the raw data for R1 and R2, which do not exhibit magnetic relaxation (Supplementary Figure 32).

Supplementary Figure 30. Repeated measurements of field-dependent longitudinal resistance for region R3 of device D1 at 1.8 K. (a,b) Field-dependent raw (a) and symmetrized (b) longitudinal resistance (R_{xx}) of recorded for region R3 of device D1 at 1.8 K. Data from six measurements, labeled 1–6, is shown. Each measurement was obtained after ZFC protocol and started with a reverse sweep (12 T to -12 T) and ended in a forward sweep (-12 T to 12 T).

Supplementary Figure 31. Repeated measurements of field-dependent longitudinal resistance for region R3 of device D1 above 1.8 K. (a–d) Raw longitudinal resistance (R_{xx}) as a function of magnetic field for region R3 at 3 K (a), 5 K (b), 10 K (c), and 20 K (d). Each dataset, labeled 1, 3, and 6, shows measurements starting with a reverse field sweep (12 T to -12 T) and ending with a forward sweep (-12 T to 12 T). The sample was held at 12 T for 5 minutes before the reverse sweep and for 60 seconds after the forward sweep.

Supplementary Figure 32. Repeated measurements of field-dependent longitudinal resistance for regions R1 and R2 of device D1 at 1.8 K. (a,b) Raw field-dependent longitudinal resistance (R_{xx}) recorded for regions R1 (a) and R2 (b) 1.8 K. Data from six measurements is shown. Each measurement started with a reverse sweep (12 T to -12 T) and ended in a forward sweep (-12 T to 12 T).

RESPONSE TO REVIEWERS' COMMENTS

Reviewer #1 (Remarks to the Author):

In their corrected version the authors responded constructively to all my questions. I recommend to accept this well done work.

We thank the reviewer for their constructive and positive feedback.

Reviewer #2 (Remarks to the Author):

Dear Editor,

I thank the Authors for their thoughtful response. I am pleased with the discussions that have emerged from my questions and the revisions made to the manuscript. Additionally, I am also positive about the way they have addressed the points raised by the other referees.

The investigation of these original magnetic lamellar materials is particularly compelling, as it offers the potential to develop devices with atomically precise interfaces for magnetic/spintronic applications. I fully support the publication of this study in your journal.

We are thankful for the reviewer's insightful and encouraging comments.

Reviewer #3 (Remarks to the Author):

The authors have provided satisfactory responses to my comments and have made the necessary adjustments to the text and figures. Therefore, I recommend the manuscript for publication.

We thank the reviewer for providing valuable feedback to improve this work.